# Evolving Multi-Scale Normalization for Time Series Forecasting under Distribution Shifts

## Abstract

Complex distribution shifts are the main obstacle to achieving accurate long-term time series forecasting. Several efforts have been conducted to capture the distribution characteristics and propose adaptive normalization techniques to alleviate the influence of distribution shifts. However, these methods neglect the intricate distribution dynamics observed from various scales and the evolving functions of distribution dynamics and normalized mapping relationships. To this end, we propose a novel model-agnostic **Evo**lving **M**ulti-**S**cale **N**ormalization (EvoMSN) framework to tackle the distribution shift problem. Flexible normalization and denormalization are proposed based on the multi-scale statistics prediction module and adaptive ensembling. An evolving optimization strategy is designed to update the forecasting model and statistics prediction module collaboratively to track the shifting distributions. We evaluate the effectiveness of EvoMSN in improving the performance of five mainstream forecasting methods on benchmark datasets and also show its superiority compared to existing advanced normalization and online learning approaches. The code is publicly available at `https://anonymous.4open.science/r/EvoMSN-4E53/`.

## 1 Introduction

Time series forecasting can provide valuable future information and thus plays an important role in many real-world applications (Lim & Zohren, 2021; Sezer et al., 2020; Qin et al., 2023; Karevan & Suykens, 2020). Abundant studies have been carried out for accurate time series forecasting, ranging from traditional statistics-based methods (Contreras et al., 2003; Holt, 2004; Theodosiou, 2011; Qiu et al., 2017) to recent deep learning-based methods (Wen et al., 2022). However, accurate forecasting of time series under distribution shifts is still a challenging and under-resolved problem.

Recently, several pioneering studies have been proposed for time series forecasting under distribution shifts by adaptive normalization approaches (Kim et al., 2021; Fan et al., 2023; Liu et al., 2024b). These approaches first remove the dynamic distribution information from the original series, enabling the model to learn from normalized data, and then predict the future distribution and recover this information to the model's output by denormalization. However, two limitations of these approaches can be identified. Firstly, existing methods model the dynamic of distribution $P(Y)$ merely from an instance level, neglecting the intricate dynamics spanning various scales. Real-world time series present diverse variations at different temporal scales, such as the electricity load exhibits unique patterns spanning hourly, daily, and weekly scales. The dynamic distribution

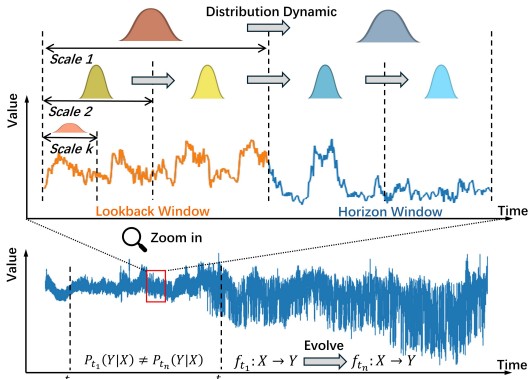

Figure 1: The marginal distribution $P(Y)$ of a time series viewed from different scales will show diverse dynamics, while the conditional distribution $P(Y|X)$ also evolves across time.

from a larger scale will show the general trend of the series, while the distribution on a smaller scale presents the local variations. The visualization of distribution dynamics from a multi-scale perspective is shown in Fig. 5 in the appendix. To this end, the modeling of statistical distribution

dynamics from a multi-scale perspective is needed but still lacking. Secondly, even though current normalization methods can help alleviate the influence of non-stationarity, they are incapable of handling changing input-output relationships caused by gradual distribution shifts, where the conditional distribution $P(Y|X)$ is not static. In this circumstance, an online approach is required for model updates with continuous data to adapt to the changing distribution over time. The problem can be illustrated by Fig. 1, where the distribution of a time series that is viewed from different scales will show diverse dynamics, and the input-output mapping function will also evolve across time.

To overcome the above limitations, we propose an **Evo**lving **M**ulti-**S**cale **N**ormalization (EvoMSN) framework for time series forecasting under complex distribution shifts. Specifically, we first propose to divide the input sequence into slices of different sizes according to their periodicity characteristics, forming views of diverse temporal scales to be normalized by the slice statistics. The backbone forecasting model can thus process the normalized series that are viewed from different scales to generate multiple outputs. Then, we propose a multi-scale statistic prediction module to capture the dynamics of statistics and predict the distributions of future slices. Consequently, we denormalize the outputs of the backbone forecasting model with the predicted distribution statistics and propose an adaptive ensemble method to aggregate the multi-scale outputs. With the proposed MSN approach, the dynamics of the distribution can be well modeled to address the non-stationarity issue. Finally, we tackle the challenge of gradual distribution shifts by proposing an evolving bi-level optimization strategy to update the statistics prediction module and the backbone forecasting model online.

In summary, our contributions are as follows:
- We propose EvoMSN, a general online normalization framework that is model-agnostic to enhance arbitrary backbone forecasting models under distribution shifts by adaptively removing and recovering the dynamical distribution information.
- We propose a multi-scale statistics prediction module to model the complex distribution dynamics and enable the estimation of future distributions. An adaptive ensembling strategy is designed in the denormalization stage to aggregate the multiple forecasting outputs and realize multi-scale modeling of time series.
- We propose an evolving bi-level optimization strategy, including offline two-stage pretraining and online alternating learning to update the statistics prediction module and the backbone forecasting model collaboratively.
- We conduct comprehensive experiments on widely used real-world benchmark datasets with various advanced forecasting methods as the backbone. Experimental results demonstrate the effectiveness of the proposed method in boosting the forecasting performance under distribution shifts and the superiority compared to other state-of-the-art normalization methods and online learning strategies.

## 2 RELATED WORKS

### 2.1 TIME SERIES FORECASTING

Traditionally, time series forecasting is carried out by utilizing statistical methods, such as the auto-regressive moving average model (Contreras et al., 2003), the exponential smoothing method (Holt, 2004), and decomposition-based methods (Theodosiou, 2011; Qiu et al., 2017). Recent decades have witnessed the fast development of deep learning-based methods to achieve accurate time series forecasting. The family of transformer models has shown great effectiveness in capturing both temporal dependency and variable dependency by applying the self-attention mechanism (Wen et al., 2022), where successful examples include Informer (Zhou et al., 2021), Autoformer (Wu et al., 2021), FEDformer (Zhou et al., 2022), Pyraformer (Liu et al., 2021), and so on. However, some recent research questions the effectiveness of transformers in time series forecasting and shows a simple linear model can achieve comparable or even superior performance (Zeng et al., 2023; Li et al., 2023; 2022). Such concerns are responded by the latest studies on transformers, which propose constructing transformers with patching approaches and showcase that transformer architecture still has its advantages in time series forecasting (Nie et al., 2022; Zhang & Yan, 2022; Chen et al., 2024). In addition to transformers, convolutional neural networks also achieve state-of-the-art performance in time series forecasting by extracting variation information along the temporal dimension (Bai et al., 2018). Multi-scale isometric convolution network (Wang et al., 2022) is proposed to learn both the local features and global correlations from time series. TimesNet (Wu et al., 2022) and

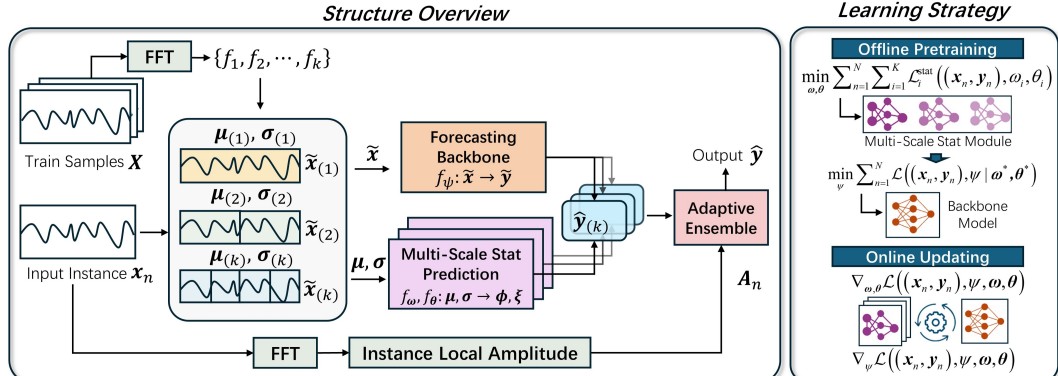

Figure 2: The proposed evolving multi-scale normalization framework. The overall structure includes periodicity extraction, multi-scale distribution dynamics modeling, and normalization-denormalization with adaptive ensembling. The learning strategy is designed with offline two-stage pretraining and online alternate updating to facilitate models to capture the evolving distribution.

PDF (Dai et al., 2023) propose to transform the 1D time series into 2D tensors based on the intrinsic periodicity characteristics, which enables the convolutional neural network to model both the intra-period and inter-period variations from a multi-scale perspective.

## 2.2 TIME SERIES FORECASTING UNDER DISTRIBUTION SHIFTS

Even though the above-mentioned methods are well-designed, they may still suffer from the complex distribution shifts problem. The inevitable temporal non-stationarity and the discrepancy between the training and testing data caused by long-term continuous distribution shifts will largely influence the forecasting model's performance and generalizability. Adaptive normalization approaches have been investigated in recent research to account for the non-stationarity issue. The deep input normalization method is proposed for time series forecasting with adaptive shifting, scaling, and shifting (Passalis et al., 2019). Kim et al. suggest an instance-level normalization and design a symmetric structure to remove and recover the distribution information according to the statistics of the instance's input window (Kim et al., 2021). Fan et al. summarize the distribution shifts problem in time series forecasting into two categories, namely intra-space shift and inter-space shift (Fan et al., 2023). They propose Dish-TS with learnable distribution coefficients for normalization and denormalization. Liu et al. further point out that the distribution shift may also happen within the window, especially in long-term tasks, and propose a finer-grained slice-based normalization strategy (Liu et al., 2024b). Liu et al. investigate the non-stationarity issue, especially for transformer models, and propose a de-stationary attention mechanism to alleviate the over-stationarization problem (Liu et al., 2022). Another class of studies mainly focuses on online learning to address the evolving distribution problem (Anava et al., 2013; Liu et al., 2016; Aydore et al., 2019), where the model will be updated with the continuous streaming data. Most recent advanced online learning approaches enhance the model adaptability under distribution shifts by complementary continual learning (Pham et al., 2022) and combining strategy (Zhang et al., 2024).

However, the above studies can only tackle one of the forms of distribution shifts but neglect scenarios where non-stationarity and evolving distribution shifts happen simultaneously, especially for online long-term time series forecasting. Besides, most existing methods incomprehensively model distribution dynamics, which only considers the statistics of the input window for normalization and denormalization. Even Liu et al. (Liu et al., 2024b) predict the distribution and carry out normalization from a finer-grained slice perspective, which is most relevant to our approach, it merely considers a static scale to model the distribution dynamics and the effectiveness is subjected to the predefined slice length. Unlike the existing methods, we propose a more comprehensive normalization framework to tackle the complex distribution shifts problem by modeling distribution dynamics in a multi-scale perspective.

## 3 PROPOSED METHOD

Given a set of $N$ multivariate time series samples with looking back windows $\boldsymbol{X} = \{\boldsymbol{x}_n\}_{n=1}^{N}$ and corresponding target horizon windows $\boldsymbol{Y} = \{\boldsymbol{y}_n\}_{n=1}^{N}$, we aim to forecast $\boldsymbol{y}_n \in \mathbb{R}^{H \times C}$ given input

$\boldsymbol{x}_n \in \mathbb{R}^{L \times C}$, where $C$ is the number of channels, $L$ and $H$ is the length of input and output window respectively.

The proposed overall framework is given as Fig. 2. Firstly, we model the distribution dynamics by periodicity extraction, multi-scale slice-based statistics calculation, and prediction. Secondly, we integrate distribution dynamics for normalization, denormalization, and adaptive ensembling of model forecasting outputs. Besides, we consider the learning procedure of the statistic prediction module and the backbone forecasting model as a bi-level optimization problem and propose a training strategy for the overall framework. Details of the proposed method are given in the following subsections.

### 3.1 MULTI-SCALE DISTRIBUTION DYNAMICS MODELING

The dynamics of distribution may exhibit diverse patterns according to different scales. For example, the daily electricity load data distribution will evolve differently from that of hourly data, where the latter distribution reflects short-term fluctuations and presents a more transient dynamic than the previous one. Considering that multi-scale modeling is usually intertwined with multi-periodicity modeling (Wu et al., 2022), we model the dynamics of distribution according to the time series periodicity properties.

**Global Periodicity Extraction**   We first analyze the time series in the frequency domain and discover the global periodicity by Fast Fourier Transform (FFT) as follows:

$$\mathbf{A} = \mathrm{Avg}\left(\mathrm{Amp}\left(\mathrm{FFT}\left(\boldsymbol{X}\right)\right)\right), \{f_1, \cdots, f_k\} = \underset{f_* \in \left\{1, \cdots, \left[\frac{t}{2}\right]\right\}}{\arg \mathrm{Topk}}\left(\mathbf{A}\right), p_i = \left\lceil \frac{t}{f_i} \right\rceil, i \in \{1, \cdots, k\}, \tag{1}$$

where $\mathrm{Amp}(\cdot)$ denotes the calculation of amplitude. The operation $\mathrm{Avg}(\cdot)$ calculates the averaged amplitude over all $M$ channels and $N$ samples to obtain $\mathbf{A} \in \mathbb{R}^t$, where $\mathbf{A}_j$ represents the intensity of the frequency-$j$ periodic basis function. $\arg \mathrm{Topk}(\cdot)$ is there to select the most prominent frequencies $\{f_1, \cdots, f_k\}$ with top-$k$ amplitudes $\{\mathbf{A}_{f_1}, \cdots, \mathbf{A}_{f_k}\}$ to represent the global periodicity with period lengths $\{p_1, \cdots, p_k\}$ $\left(p_i = \left\lceil \frac{t}{f_i} \right\rceil\right)$.

**Multi-Scale Slice-based Statistics Calculation and Prediction**   Based on the discovered global periodicity, we propose to split looking back windows and horizon windows into non-overlapping slices with respect to different period lengths $\{p_1, \cdots, p_k\}$ as follows:

$$\boldsymbol{X}_{(i)} = \mathrm{Slicing}_{p_i}\left(\mathrm{Padding}\left(\boldsymbol{X}\right)\right), \boldsymbol{Y}_{(i)} = \mathrm{Slicing}_{p_i}\left(\mathrm{Padding}\left(\boldsymbol{Y}\right)\right), i \in \{1, \cdots, k\}, \tag{2}$$

where $\mathrm{Padding}(\cdot)$ is to first extend the time series by copying a segment of itself from the end to make it compatible for $\mathrm{Slicing}_{p_i}(\cdot)$ with length $p_i$. Then, the slicing process will transform the original windows into a set of sliced windows $\left\{\boldsymbol{X}_{(1)}, \cdots, \boldsymbol{X}_{(k)}\right\}$ and $\left\{\boldsymbol{Y}_{(1)}, \cdots, \boldsymbol{Y}_{(k)}\right\}$ with respect to different periods, where $\boldsymbol{X}_{(i)} = \left\{\left\{\boldsymbol{x}_{(i),n}^j\right\}_{j=1}^{\frac{L}{p_i}}\right\}_{n=1}^N$, $\boldsymbol{X}_{(i)} \in \mathbb{R}^{N \times \frac{L}{p_i} \times p_i \times C}$ and $\boldsymbol{Y}_{(i)} = \left\{\left\{\boldsymbol{y}_{(i),n}^j\right\}_{j=1}^{\frac{H}{p_i}}\right\}_{n=1}^N$, $\boldsymbol{Y}_{(i)} \in \mathbb{R}^{N \times \frac{H}{p_i} \times p_i \times C}$. Then, the mean and standard deviation for each slice are computed as:

$$\mu_{(i),n}^j = \frac{1}{p_i} \sum_{t=1}^{p_i} \boldsymbol{x}_{(i),n}^{j,t}, \quad \left(\sigma_{(i),n}^j\right)^2 = \frac{1}{p_i} \sum_{t=1}^{p_i} \left(\boldsymbol{x}_{(i),n}^{j,t} - \mu_{(i),n}^j\right)^2, \tag{3}$$

$$\phi_{(i),n}^j = \frac{1}{p_i} \sum_{t=1}^{p_i} \boldsymbol{y}_{(i),n}^{j,t}, \quad \left(\xi_{(i),n}^j\right)^2 = \frac{1}{p_i} \sum_{t=1}^{p_i} \left(\boldsymbol{y}_{(i),n}^{j,t} - \phi_{(i),n}^j\right)^2, \tag{4}$$

where $\mu_{(i),n}^j, \sigma_{(i),n}^j, \phi_{(i),n}^j, \xi_{(i),n}^j \in \mathbb{R}^{1 \times M}$, $\mu_{(i),n}^j, \sigma_{(i),n}^j$ are mean and standard deviation of slice $\boldsymbol{x}_{(i),n}^j$, and $\phi_{(i),n}^j, \xi_{(i),n}^j$ are statistics of $\boldsymbol{y}_{(i),n}^j$.

Considering the multi-scale characteristics of time series data with corresponding evolving distribution, we propose a multi-scale prediction module to capture the dynamics of statistics and predict the distributions of future slices. Concretely, the dynamics of mean and standard deviation are modeled separately, and specific models for each periodicity are constructed:

$$\hat{\boldsymbol{\phi}}_{(i)} = f_{\omega_i}\left(\boldsymbol{\mu}_{(i)}, \boldsymbol{x}\right), \quad \hat{\boldsymbol{\xi}}_{(i)} = f_{\theta_i}\left(\boldsymbol{\sigma}_{(i)}, \boldsymbol{x}\right), \tag{5}$$

where $f_{\omega_i}(\cdot)$ and $f_{\theta_i}(\cdot)$ are prediction models parameterized by $\omega_i$ and $\theta_i$ for modeling mean and standard deviation statistics under periodicity-$i$. Each prediction model is designed to be lightweight and consists of a two-layer perceptron network with $\mathrm{Relu}()$ and $\mathrm{Identity}()$ as final activations for standard deviation and mean statistics, respectively.

## 3.2 Normalization with Adaptive Ensemble

Our proposed method alleviates the influence of non-stationarity on the forecasting model by following the logic of "normalization - backbone model forecasting - denormalization" as existing normalization methods (Kim et al., 2021; Fan et al., 2023; Liu et al., 2024b). Moreover, the proposed multi-scale statistics prediction module can provide more comprehensive information about how data statistics are evolving. To this end, our method will tackle the distribution shift challenge from a multi-scale perspective and further improve forecasting by adaptive ensembling.

**Normalization** Firstly, we consider the periodicity with intricate distribution shifts of a given series $\boldsymbol{x}_n$ and normalize it with slice statistics:

$$\tilde{\boldsymbol{x}}_{(i),n}^j = \frac{1}{\sigma_{(i),n}^j + \varepsilon} * \left( \boldsymbol{x}_{(i),n}^j - \mu_{(i),n}^j \right), \tag{6}$$

where $\tilde{\boldsymbol{x}}_{(i),n}^j$ is the normalized series corresponding to slicing with periodicity-$i$, $\varepsilon$ is a small constant value to prevent calculation instability. The operation * denotes the per-element product. To this end, a set of $k$ normalized series can be obtained $\tilde{\boldsymbol{x}}_n = \left\{ \tilde{\boldsymbol{x}}_{(i),n} \right\}_{i=1}^k$, which ensures each period of series has a similar statistics and thus to be easier analyzed by the backbone forecasting model.

**Denormalization** Subsequently, the set of normalized series will be processed by an arbitrary backbone forecasting model $f_\psi(\cdot)$:

$$\tilde{\boldsymbol{y}}_n = \left\{ \tilde{\boldsymbol{y}}_{(i),n} \right\}_{i=1}^k = f_\psi(\tilde{\boldsymbol{x}}_n), \tag{7}$$

where $\tilde{\boldsymbol{y}}_n$ is a set of $k$ normalized output generated by the backbone model with the same set of parameter $\psi$. These outputs will then be split into slices and denormalized by the predicted slice statistics given by the multi-period statistics prediction module to recover the non-stationary information of different periodicities:

$$\hat{\boldsymbol{y}}_{(i),n}^j = \tilde{\boldsymbol{y}}_{(i),n}^j * \left( \hat{\xi}_{(i),n}^j + \varepsilon \right) + \hat{\phi}_{(i),n}^j. \tag{8}$$

By concatenating the denormalized slices in their chronological order, we can obtain a set of output series $\left\{ \hat{\boldsymbol{y}}_{(i),n} \right\}_{i=1}^k$.

**Multi-Scale Adaptive Ensemble** Even though the periodicity is analyzed from a global perspective where the dominant periodicities are determined by the average amplitude as equation 1, each individual series and channel may show different degrees of these periodicities. With this consideration, we analyze the periodicity from a local perspective by calculating the local amplitude of the global periodicity for a given input series $\boldsymbol{x}_n$ as:

$$\boldsymbol{A}_n = \underset{\{f_1, \cdots, f_k\}}{\mathrm{Amp}} \left( \mathrm{FFT}(\boldsymbol{x}_n) \right), \tag{9}$$

where $\boldsymbol{A}_n \in \mathbb{R}^{k \times C}$. As the amplitudes can reflect the relative importance of the periodicity, we adaptively ensemble the denormalized outputs based on the weights of the local amplitude as:

$$\hat{\boldsymbol{y}}_n = \sum \boldsymbol{w}_{(i),n} * \hat{\boldsymbol{y}}_{(i),n}, \quad w_{(i),n} = \frac{A_{(i),n}}{\sum_{i=1}^k A_{(i),n}} \tag{10}$$

## 3.3 Evolving Bi-Level Optimization

The performance of the backbone forecasting model is subject to the output given by the statistics prediction module. To this end, the training of these models is formulated as a bi-level optimization problem as follows:

$$\min_{\psi} \sum_{n=1}^N \mathcal{L}\left( (\boldsymbol{x}_n, \boldsymbol{y}_n), \psi | \boldsymbol{\omega}^*, \boldsymbol{\theta}^* \right)$$
$$\text{s.t. } \boldsymbol{\omega}^*, \boldsymbol{\theta}^* = \arg\min_{\boldsymbol{\omega}, \boldsymbol{\theta}} \sum_{n=1}^N \sum_{i=1}^k \mathcal{L}_i^{stat}\left( (\boldsymbol{x}_n, \boldsymbol{y}_n), \psi, \boldsymbol{\omega}_i, \boldsymbol{\theta}_i \right), \tag{11}$$

where the upper objective is to optimize the backbone forecasting model with the normal MSE loss function $\mathcal{L}$ and the lower objective is to minimize the distribution discrepancy between the denormalized output $\hat{\boldsymbol{y}}_{(i)} = f(\boldsymbol{x}_n, \psi, \omega_i, \theta_i)$ and true distribution of $\boldsymbol{y}_n$ in views of different scales.

To solve such a bi-level optimization problem, we design a training strategy including offline decoupled pretraining and online alternate updating. In the offline pretraining stage, we first decoupled the optimization into two separate processes to enable the multi-scale statistics prediction module to focus on estimating the future distributions and let the backbone model focus on learning from normalized series generated by statistics of different scales. Concretely, the statistics prediction module is trained by optimizing $\boldsymbol{\omega}^*, \boldsymbol{\theta}^* = \arg\min_{\boldsymbol{\omega},\boldsymbol{\theta}} \sum_{n=1}^{N} \sum_{i=1}^{k} \mathcal{L}_i^{\text{stat}} ((\boldsymbol{x}_n, \boldsymbol{y}_n), \omega_i, \theta_i)$, which mainly focuses on estimating future statistics information regardless of the backbone forecasting model. With such simplification, the original challenging task of minimizing the distribution discrepancy between two series is transformed to minimize the difference between the estimated slice-based statistics and the corresponding ground truth. To this end, The statistics prediction module corresponding to periodicity-$i$ can be trained with the loss function calculated by the mean squared error $\ell((\hat{\boldsymbol{\phi}}_{(i)}, \hat{\boldsymbol{\xi}}_{(i)}), (\boldsymbol{\phi}_{(i)}, \boldsymbol{\xi}_{(i)}))$. After the statistics prediction module is trained to converge, the backbone forecasting model is optimized to minimize the loss between the ensembled multi-scale output and the ground truth value $\ell(\hat{\boldsymbol{y}}_n, \boldsymbol{y}_n)$.

Considering the evolving characteristics of both distribution dynamics and the forecasting model's input-output relationship, an alternate updating strategy is then designed to enable the model to learn from continuous data samples online. In the online learning stage, the forecasting loss between the denormalized series and the ground truth is set as the overall optimization target for both the backbone model and the multi-scale statistics prediction module, which enables a tighter collaboration between these components. We first freeze the backbone forecasting model and update the multi-scale statistics prediction module by descending $\nabla_{\boldsymbol{\omega},\boldsymbol{\theta}} \mathcal{L} ((\boldsymbol{x}_n, \boldsymbol{y}_n), \psi, \boldsymbol{\omega}, \boldsymbol{\theta})$. For the next coming data, we freeze the statistics prediction module as the condition for updating the backbone forecasting model with gradient $\nabla_{\psi} \mathcal{L} ((\boldsymbol{x}_n, \boldsymbol{y}_n), \psi, \boldsymbol{\omega}, \boldsymbol{\theta})$. We repeat such alternating updates with online streaming data to enable both the statistics prediction module and the backbone model to track the evolving distribution.

# 4 EXPERIMENTS

## 4.1 EXPERIMENTAL SETUP

**Datasets** We evaluate our methods on five large-scale real-world time-series datasets: (1) **Electricity transformer temperature (ETT)** [1] records of oil temperature and electricity transformers' power load from July 2016 to July 2018. We choose the hourly ETTh1 data and 15-minute-resolution Ettm1 data for our experiments. (2) **Electricity** [2] contains hourly electricity load data of 321 clients from July 2016 to July 2019. (3) **Exchange** [3] collects the panel data of daily exchange rates from 8 countries from 1990 to 2016. (4) **Traffic**[4] contains hourly traffic load recorded by 862 sensors from January 2015 to December 2016. (5) **Weather**[5] contains meteorological time series with 21 weather indicators collected every 10 minutes in 2020.

**Backbone Models** We evaluate the proposed model-agnostic EvoMSN framework with various mainstream time series models as the backbone under both online and offline multivariate forecasting settings. We selected backbone models including linear model **DLinear** (Zeng et al., 2023), transformers **Autoformer** (Wu et al., 2021), **FEDformer** (Zhou et al., 2022), **PatchTST** (Nie et al., 2022), and convolutional model **TimesNet** (Wu et al., 2022).

**Implementation Settings** We evaluate the effectiveness of the proposed method in tackling the distribution shifts problem under two scenarios, including online and offline forecasting. For online forecasting, we split the data into warm-up pretraining and online learning phases by the ratio of 20:75. We follow the optimization details in (Pham et al., 2022) by optimizing the $l_2$ (MSE) loss with the AdamW optimizer (Loshchilov & Hutter, 2017). According to the widely applied online setting, the forecasts will be made as each test data sample arrives, and the model will be updated by one epoch according to the online forecasting loss. For offline forecasting, we split the training and testing data with the ratio of 70:20 and follow the implementation details in (Liu et al., 2024b). The hyperparameter tuning is conducted based on the performance of a separate validation set. We utilize the mean squared error (MSE) and mean absolute error (MAE) as the metrics to evaluate

---

[1] https://github.com/zhouhaoyi/ETDataset
[2] https://archive.ics.uci.edu/ml/datasets/ElectricityLoadDiagrams20112014
[3] https://github.com/laiguokun/multivariate-time-series-data
[4] http://pems.dot.ca.gov
[5] https://www.bgc-jena.mpg.de/wetter/

forecasting performance. All the experiments are conducted with PyTorch (Paszke et al., 2019) on a single NVIDIA GeForce RTX 3080 Ti 12GB GPU.

## 4.2 MAIN RESULTS

Table 1: Online multivariate forecasting results. The **bold** values indicate better performance when the backbone model is equipped with the proposed EvoMSN method.

| Methods | | Autoformer | | +EvoMSN | | FEDformer | | +EvoMSN | | Dlinear | | +EvoMSN | | PatchTST | | +EvoMSN | | TimesNet | | +EvoMSN |
|---|---|---|---|---|---|---|---|---|---|---|---|---|---|---|---|---|---|---|---|---|
| Metric | | MSE | MAE | MSE | MAE | MSE | MAE | MSE | MAE | MSE | MAE | MSE | MAE | MSE | MAE | MSE | MAE | MSE | MAE | MSE | MAE |
| Exchange | 96 | 0.193 | 0.261 | **0.136** | **0.223** | 0.171 | 0.248 | **0.080** | **0.162** | 0.131 | 0.252 | **0.119** | **0.224** | 0.110 | 0.213 | **0.093** | **0.202** | 0.050 | 0.129 | 0.054 | 0.143 |
| | 192 | 0.282 | 0.293 | **0.181** | **0.249** | 0.205 | 0.267 | **0.124** | **0.189** | 0.225 | 0.334 | **0.160** | **0.270** | 0.148 | 0.255 | **0.125** | **0.237** | 0.067 | 0.159 | **0.043** | **0.133** |
| | 336 | 0.291 | 0.340 | **0.227** | **0.296** | 0.267 | 0.311 | **0.120** | **0.212** | 0.373 | 0.428 | **0.220** | **0.320** | 0.197 | 0.296 | **0.169** | **0.273** | 0.096 | 0.195 | **0.091** | **0.181** |
| Traffic | 96 | 0.621 | 0.478 | **0.219** | **0.249** | 0.573 | 0.447 | **0.228** | **0.244** | 0.659 | 0.395 | **0.380** | **0.288** | 0.353 | 0.304 | **0.317** | **0.269** | 0.301 | 0.274 | **0.219** | **0.240** |
| | 192 | 0.617 | 0.471 | **0.253** | **0.256** | 0.324 | 0.346 | **0.226** | **0.245** | 0.604 | 0.370 | **0.396** | **0.294** | 0.344 | 0.301 | **0.326** | **0.275** | 0.234 | 0.235 | **0.230** | 0.240 |
| | 336 | 0.553 | 0.436 | **0.269** | **0.259** | 0.283 | 0.315 | **0.227** | **0.243** | 0.581 | 0.395 | **0.413** | **0.304** | 0.357 | 0.300 | **0.337** | **0.280** | 0.291 | 0.258 | **0.272** | 0.263 |
| Electricity | 96 | 3.007 | 0.401 | **1.358** | **0.286** | 2.533 | 0.762 | **1.608** | **0.279** | 3.702 | 0.372 | 6.840 | 0.389 | 4.451 | 0.548 | **3.701** | **0.405** | 2.038 | 0.282 | **1.684** | **0.280** |
| | 192 | 3.575 | 0.429 | **2.096** | **0.319** | 2.034 | 0.635 | **1.662** | **0.274** | 4.787 | 0.401 | 7.518 | 0.414 | 4.550 | 0.571 | 4.710 | **0.435** | 2.485 | 0.305 | **1.599** | **0.288** |
| | 336 | 4.110 | 0.430 | **2.390** | **0.321** | 1.945 | 0.589 | 1.962 | **0.282** | 10.692 | 0.449 | **8.533** | **0.437** | 5.371 | 0.601 | 5.941 | **0.494** | 3.488 | 0.343 | **2.186** | **0.312** |
| Weather | 96 | 1.702 | 0.459 | **0.973** | 0.530 | 0.533 | 0.345 | 0.586 | 0.353 | 1.001 | 0.470 | **0.669** | **0.395** | 0.796 | 0.427 | **0.728** | **0.414** | 0.699 | 0.359 | **0.407** | **0.237** |
| | 192 | 1.481 | 0.497 | **1.013** | 0.562 | 0.518 | 0.346 | 0.738 | 0.434 | 1.141 | 0.528 | **0.744** | **0.437** | 0.691 | 0.421 | **0.661** | **0.411** | 0.831 | 0.403 | **0.413** | **0.247** |
| | 336 | 1.473 | 0.548 | **1.005** | 0.566 | 0.487 | 0.352 | 0.771 | 0.448 | 1.276 | 0.582 | **0.819** | **0.473** | 0.708 | 0.438 | **0.687** | **0.424** | 1.158 | 0.441 | **0.387** | **0.237** |
| ETTh1 | 96 | 0.534 | 0.476 | **0.291** | **0.360** | 0.416 | 0.454 | **0.379** | **0.418** | 0.777 | 0.558 | **0.640** | **0.533** | 0.512 | 0.509 | **0.487** | **0.481** | 0.469 | 0.435 | **0.314** | **0.372** |
| | 192 | 0.556 | 0.485 | **0.475** | **0.457** | 0.416 | 0.453 | **0.397** | **0.427** | 0.869 | 0.598 | **0.659** | **0.554** | 0.526 | 0.516 | 0.562 | **0.515** | 0.419 | 0.412 | **0.403** | 0.424 |
| | 336 | 0.634 | 0.525 | **0.528** | **0.494** | 0.400 | 0.435 | **0.397** | 0.437 | 0.930 | 0.628 | **0.757** | **0.595** | 0.541 | 0.527 | 0.588 | **0.526** | 0.497 | 0.451 | **0.274** | **0.353** |
| ETTm1 | 96 | 0.256 | 0.351 | **0.172** | **0.300** | 0.185 | 0.317 | **0.169** | **0.288** | 0.391 | 0.449 | **0.268** | **0.379** | 0.272 | 0.364 | **0.199** | **0.329** | 0.122 | 0.250 | **0.109** | **0.236** |
| | 192 | 0.300 | 0.377 | **0.193** | **0.323** | 0.231 | 0.313 | **0.153** | **0.276** | 0.455 | 0.482 | **0.320** | **0.410** | 0.265 | 0.365 | **0.204** | **0.334** | 0.120 | 0.248 | **0.101** | **0.227** |
| | 336 | 0.292 | 0.393 | **0.267** | **0.379** | 0.211 | 0.329 | 0.255 | 0.362 | 0.544 | 0.523 | **0.386** | **0.446** | 0.284 | 0.376 | **0.237** | **0.350** | 0.114 | 0.239 | **0.102** | **0.230** |

We report the online multivariate forecasting results in Table 1. The length of the online long-term forecasting window is set as $L_{\text{out}} \in \{96, 192, 336\}$. We set the length of the input sequence according to the original setting of the backbone forecasting model as $L_{\text{in}} = 96$ for Autoformer, FEDformer, TimesNet, and $L_{\text{in}} = 336$ for DLinear and PatchTST.

For the online forecasting, we set the baseline as letting the backbone models pretrain on the training data set and adopt simple online learning strategies to train and test continuously on the testing data. The proposed EvoMSN framework enables the fine-grained modeling of distribution dynamics and time series evolving patterns, which greatly enhances the forecasting performance of most benchmark models across various forecasting horizons. Taking TimesNet as an example, EvoMSN can achieve an average reduction in MSE by **11.30%** on the Exchange dataset, **12.77%** on the Traffic dataset, **31.74%** on the Electricity dataset, **55.09%** on the Weather dataset, **28.41%** on the ETTh1 dataset, and **12.12%** on the ETTm1 dataset. The results also demonstrate that the proposed method can improve the performance of transformers and linear model across different forecasting horizons. It should be noted that even though PatchTST and TimesNet involve similar concepts of slicing and multi-scale modeling in their model design, these methods still benefit from the proposed normalization framework because the decoupled and explicit modeling of distribution dynamics enables these backbones to better learn from a stationarized series. For a better visualization, we show the online long-term forecasting results with the output window length of 336 in Figure 3. The results are given by setting DLinear as the backbone model, where the vanilla online learning strategy and the proposed EvoMSN method are compared. We can observe that the model is struggling to track the complex evolving distributions with the vanilla online learning strategy, while the proposed EvoMSN method adaptively adjusts the distribution of forecasting outputs to align with the ground truth. More visualization results can be found in Appendix A.1.

To further validate the efficacy of the proposed method, we have analyzed its performance on large-scale data, low-frequency data, and computation efficiency, where detailed results can be found in Appendix A.2. We have also conducted comprehensive experiments to showcase the effectiveness of the proposed method in enhancing offline long-term non-stationary time series forecasting, where the results are provided in Appendix A.5. The results show that the proposed method can also help tackle the distribution shift problem without the online learning process.

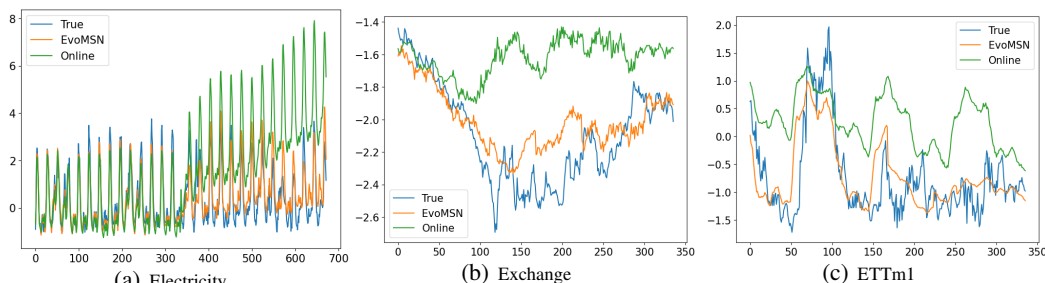

(a) Electricity       (b) Exchange       (c) ETTm1

Figure 3: Visualization of online long-term forecasting results with the output window length of 336.

### 4.3 COMPARISON WITH ONLINE LEARNING STRATEGIES

In order to showcase the effectiveness of the proposed method, we also compare it with other advanced online learning strategies that all use Temporal Convolution Network (TCN) (Bai et al., 2018) as the backbone. First, the **Online-TCN** adopts the vanilla online learning strategy to train continuously with the streaming data. Then, we consider the Experience Replay (**ER**) strategy (Chaudhry et al., 2019) that employs a buffer to store previous data and update with new coming data. **DER++** (Buzzega et al., 2020) is the variant of ER, which augments the standard ER with a knowledge distillation strategy (Hinton et al., 2015). Lastly, we consider the previous state-of-the-art online learning method **FSNet** (Pham et al., 2022), which adopts a complementary continual learning strategy. We use the same setting for these methods for a fair comparison, where the input window length is 96, and the learning rate is 0.001. We equip the proposed EvoMSN method to TCN (denoted as **EvoMSN-TCN**) and report the comparison results with existing online strategies in Table 2. We can observe that ER, DER++, and FSNet are strong competitors and can effectively

Table 2: Comparison between EvoMSN and existing online learning strategies. The best and second best results are in **bold** and underline.

| Methods | EvoMSN-TCN | | Online-TCN | | FSNet | | ER | | DER++ | |
|---|---|---|---|---|---|---|---|---|---|---|
| Metric | MSE | MAE | MSE | MAE | MSE | MAE | MSE | MAE | MSE | MAE |
| Exchange | **0.095** | **0.176** | 0.244 | 0.319 | 0.259 | 0.333 | 0.224 | 0.311 | 0.230 | 0.318 |
| Traffic | **0.486** | **0.374** | 0.849 | 0.513 | 0.944 | 0.562 | 0.837 | 0.508 | 0.842 | 0.508 |
| Electricity | **4.122** | **0.397** | 15.721 | 1.215 | 14.109 | 1.025 | 14.790 | 1.207 | 14.926 | 1.146 |
| Weather | **0.734** | **0.421** | 0.822 | 0.484 | 0.923 | 0.544 | 0.760 | 0.452 | 0.773 | 0.459 |
| ETTh1 | **0.669** | **0.555** | 0.776 | 0.589 | 0.675 | 0.571 | 0.766 | 0.588 | 0.721 | 0.570 |
| ETTm1 | **0.323** | **0.423** | 0.447 | 0.507 | 0.473 | 0.521 | 0.430 | 0.496 | 0.430 | 0.495 |

improve the online forecasting performance compared to the vanilla method. However, such methods do not model the dynamics of distribution and cannot fully address the complex distribution shift problem in the presence of both non-stationarity and gradual distribution shifts. Especially for FSNet, which performs well in the original work that focuses on short-term forecasting but degrades with a longer forecasting horizon. In comparison, the proposed EvoMSN method largely improves the capability of the TCN model in the online setting and shows promising results on all datasets that outperform the strong baselines. The EvoMSN-TCN can achieve a performance improvement that is measured on all datasets across all forecasting horizons by **65.90%** compared to Online-TCN, **63.89%** compared to ER, **64.12%** compared to DER++, and **63.01%** compared to FSNet. To investigate the model performance during the online forecasting process, we plot the cumulative average MSE loss of different online methods in Fig 4. With such visualization, the distribution shifts are likely to happen when the cumulative loss increases or shows peaks. The results clearly show the EvoMSN-TCN has a much lower and smoother cumulative loss curve, especially for challenging Electricity, Exchange, and Traffic datasets that are highly non-stationary. The above observations validate that the proposed EvoMSN method can effectively enhance the model for online forecasting, where comprehensive results are provided in Appendix A.3.

### 4.4 COMPARISON WITH NORMALIZATION METHODS

Subsequently, we compare EvoMSN with three state-of-the-art normalization methods for time series forecasting under distribution shifts: **SAN** (Liu et al., 2024b), **RevIN** (Kim et al., 2021), and **Dish-TS** (Fan et al., 2023). As these normalization methods were originally proposed for the of-

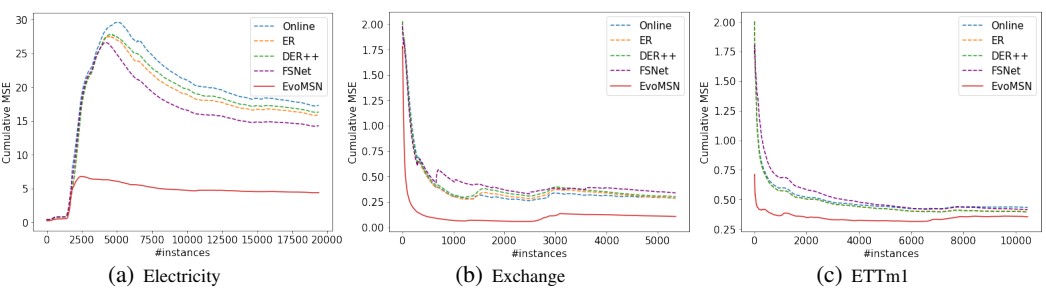

(a) Electricity          (b) Exchange          (c) ETTm1

Figure 4: Evolution of the cumulative average MSE loss during online forecasting.

Table 3: Comparison between EvoMSN and existing normalization approaches for online forecasting.

| Methods | Autoformer | | | | | | | | TimesNet | | | | | | | |
|---|---|---|---|---|---|---|---|---|---|---|---|---|---|---|---|---|
| | +EvoMSN | | +SAN | | +RevIN | | +Dish-TS | | +EvoMSN | | +SAN | | +RevIN | | +Dish-TS | |
| Metric | MSE | MAE | MSE | MAE | MSE | MAE | MSE | MAE | MSE | MAE | MSE | MAE | MSE | MAE | MSE | MAE |
| Exchange | **0.181** | 0.256 | 0.219 | 0.265 | 0.193 | **0.236** | 0.231 | 0.248 | **0.063** | **0.153** | 0.114 | 0.184 | 0.071 | 0.161 | 0.065 | 0.164 |
| Traffic | **0.247** | **0.255** | 0.304 | 0.287 | 0.351 | 0.309 | 0.342 | 0.307 | **0.240** | **0.248** | 0.255 | 0.254 | 0.275 | 0.256 | 0.379 | 0.313 |
| Electricity | **1.948** | **0.308** | 2.965 | 0.341 | 3.179 | 0.380 | 3.536 | 0.411 | **1.823** | **0.293** | 2.480 | 0.313 | 2.670 | 0.310 | 3.315 | 0.384 |
| Weather | **0.997** | 0.553 | 1.187 | **0.502** | 1.154 | 0.560 | 1.062 | 0.539 | **0.402** | **0.240** | 0.719 | 0.385 | 0.896 | 0.401 | 0.737 | 0.420 |
| ETTh1 | **0.431** | **0.437** | 0.529 | 0.481 | 0.484 | 0.464 | 0.517 | 0.482 | **0.330** | **0.383** | 0.379 | 0.410 | 0.462 | 0.433 | 0.448 | 0.439 |
| ETTm1 | **0.211** | **0.334** | 0.247 | 0.362 | 0.315 | 0.414 | 0.257 | 0.375 | **0.104** | **0.231** | 0.138 | 0.269 | 0.119 | 0.246 | 0.169 | 0.299 |

fline forecasting setting, we apply them to online forecasting by updating the backbone model with streaming data. The comparison of normalization approaches for online forecasting based on Autoformer and TimesNet backbones are presented in Table 3. It can be concluded that EvoMSN achieves the best performance among existing normalization methods. The RevIN and Dish-TS mainly utilize statistics of the whole input window for normalization and denormalization; such a coarse way causes them a relatively inferior performance. SAN adopts a finer slice-based approach for normalization and explicitly models the distribution dynamics to predict future distribution for denormalization. To this end, SAN achieves a better performance than RevIN and Dish-TS. However, SAN only considers a single scale to model the distribution dynamics but neglects the difference in distribution dynamics across various scales. Besides, SAN only considers the offline two-stage training scheme but is not tailored for online forecasting. Distinct from these methods, the proposed EvoMSN adopts a more comprehensive multi-scale approach to capture the evolving distribution and consequently achieves the best performance, where detailed results across various forecasting horizons are provided in Appendix A.4. In addition, we compare forecasting performance when different normalization approaches are combined with the SOTA continual learning approach (FSNet) and show the consistent advantage of EvoMSN in Appendix A.4. The superiority of the proposed method versus other normalization methods is also validated for offline long-term forecasting, where results are provided in Appendix A.6.

## 4.5 ABALTION STUDY

This section investigates the impact of the multi-scale modeling approach and the proposed evolving optimization strategy, where the DLinear model is utilized as the backbone. First, we vary the number of scales in modeling multi-scale distribution dynamics and report results in Table 4. We find an overall trend that the model performance is better when more scales are considered to model the distribution dynamics, where $K = 4$ achieves the best performance and is set as default in our experiments. Compared to considering a single scale, multi-scale modeling approaches can reduce the average MSE loss by **23.82%** on Exchange and **12.64%** on ETTm1, which validates the necessity of the proposed multi-scale modeling strategy. Then, we investigate the proposed evolving updation strategy with different ablation settings in Table 5: W/O online means only carrying out offline two-stage pretraining without an online updating of both the backbone model and statistics prediction module. W/O stat and W/O backbone mean online learning without an update of the backbone model and statistics prediction module, respectively. We can find that only modeling the distribution dynamics in the offline pretraining process is inadequate to tackle the online forecasting challenges, where the functions of both distribution dynamics and normalized input-output relationship will evolve over time. To this end, the online updation of the statistics prediction module and

Table 4: Sensitivity study. The online prediction accuracy varies with the number of multi-scales $K$.

| Number of Scales | | $K=1$ | | $K=2$ | | $K=3$ | | $K=4$ | |
|---|---|---|---|---|---|---|---|---|---|
| Metric | | MSE | MAE | MSE | MAE | MSE | MAE | MSE | MAE |
| | 96 | 0.159 | 0.261 | 0.148 | 0.249 | 0.131 | 0.237 | **0.119** | **0.224** |
| Exchange | 192 | 0.198 | 0.302 | 0.241 | 0.324 | 0.190 | 0.290 | **0.160** | **0.270** |
| | 336 | 0.298 | 0.384 | 0.310 | 0.379 | 0.246 | 0.343 | **0.220** | **0.320** |
| | 96 | 0.283 | 0.387 | 0.298 | 0.399 | 0.282 | 0.389 | **0.268** | **0.379** |
| ETTm1 | 192 | 0.360 | 0.437 | 0.334 | 0.421 | 0.322 | 0.412 | **0.320** | **0.410** |
| | 336 | 0.473 | 0.501 | 0.401 | 0.457 | 0.421 | 0.468 | **0.386** | **0.446** |

Table 5: Ablation study. W/O online, W/O stat, and W/O backbone represent forecasting without online updating of both the backbone forecasting model and the statistics prediction module, without online updating of the statistics prediction module, and without online updating of the backbone forecasting model, respectively.

| Models | | W/O online | | W/O stat | | W/O backbone | | EvoMSN | |
|---|---|---|---|---|---|---|---|---|---|
| Metric | | MSE | MAE | MSE | MAE | MSE | MAE | MSE | MAE |
| | 96 | 0.413 | 0.390 | 0.355 | 0.361 | 0.305 | 0.356 | **0.119** | **0.224** |
| Exchange | 192 | 0.741 | 0.529 | 0.632 | 0.491 | 0.531 | 0.483 | **0.160** | **0.270** |
| | 336 | 1.203 | 0.693 | 1.110 | 0.663 | 0.898 | 0.628 | **0.220** | **0.320** |
| | 96 | 0.561 | 0.539 | 0.508 | 0.501 | 0.528 | 0.527 | **0.268** | **0.379** |
| ETTm1 | 192 | 0.660 | 0.576 | 0.598 | 0.533 | 0.623 | 0.565 | **0.320** | **0.410** |
| | 336 | 0.798 | 0.623 | 0.725 | 0.582 | 0.750 | 0.611 | **0.386** | **0.446** |

backbone forecasting model are both important to address the distribution shifts problem, where the proposed online alternating updation strategy provides a promising solution.

## 5 CONCLUSION

In this paper, we propose an evolving multi-scale normalization framework for time series forecasting under complex distribution shifts, which enables a more comprehensive way to model the distribution dynamics and facilitates the forecasting model to better track the evolving distributions. As a model-agnostic framework with arbitrary backbones, the proposed method effectively boosts mainstream forecasting techniques to achieve state-of-the-art performance on real-world time-series datasets by a significant margin. Extensive experiments have been conducted to examine the superiority of the proposed method in tackling distribution shift challenges compared to other advanced normalization and online learning strategies. As our future direction, we will further investigate a more general normalization framework that takes into account more detailed distribution information beyond mean and variance.

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

# A SUPPLEMENTARY EXPERIMENTS

## A.1 FULL VISUALIZATION RESULTS

The methodology designed in this work is motivated by the fact that time series data distribution shows diverse dynamics across different scales. We plot the statistics (mean and standard deviation) of windows that have different scales in Fig. 5. We can see from the figure that a larger scale shows the general trend of how the distribution is evolving while a smaller scale presents more detailed local variations.

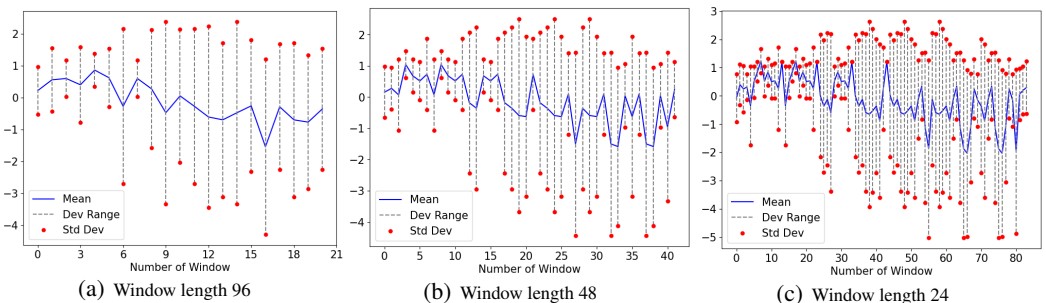

(a) Window length 96     (b) Window length 48     (c) Window length 24

Figure 5: Visualization of statistics of windows. The blue line represents the mean of each window. Red dots represent the window mean plus/minus the window standard deviation, and the gray dash line represents the deviation range of the window. (a), (b), (c) plots the statistics of each window when the window length equals to 96, 48, and 24, respectively.

The visualization results in comparing the forecasting with and without the proposed EvoMSN method are shown in Fig. 6, which reports the performance of the Dlinear backbone on Electricity, Exchange, Traffic, Weather, ETTh1, and ETTm1 data. It shows clearly that the vanilla online learning strategy is inadequate to address the complex distribution shift problem, where the proposed EvoMSN approach is of great necessity to enable the model to generate outputs with a more reasonable distribution.

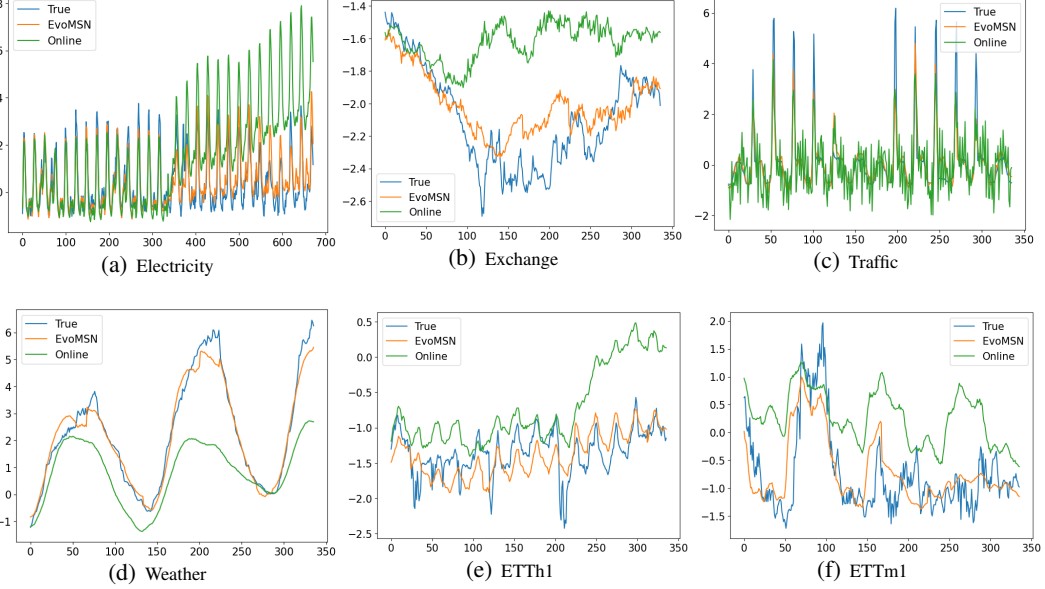

(a) Electricity     (b) Exchange     (c) Traffic

(d) Weather     (e) ETTh1     (f) ETTm1

Figure 6: Visualization of online long-term forecasting results with the output window length of 336. The results are given by setting DLinear as the backbone model, where the vanilla online learning strategy and the proposed EvoMSN method are compared.

## A.2 EXTENDED EXPERIMENTS RESULTS

We conduct experiments on a larger traffic dataset (Liu et al., 2024a) to online forecast hourly traffic flow with 716 sensors in San Diego county in 2019. The proposed EvoMSN method is applied to three competing backbone forecasting models in the reference, including **LSTM** (Hochreiter, 1997), **STGCN** (Yu et al., 2017), and **GWNET** (Wu et al., 2019). The results are shown in Table 6, where EvoMSN can achieve an average reduction in RMSE by **18.94%** with LSTM, **10.08%** with STGCN, and **7.3%** with GWNET compared to the vanilla online approach.

Table 6: Online forecasting results on large traffic dataset.

|  | | Pred_len 48 | | Pred_len 72 | | Pred_len 96 | | Avg | |
|---|---|---|---|---|---|---|---|---|---|
|  | | MAE | RMSE | MAE | RMSE | MAE | RMSE | MAE | RMSE |
| Online LSTM | | 52.146 | 76.754 | 52.538 | 77.442 | 53.347 | 78.312 | 52.677 | 77.503 |
| | **+EvoMSN** | **41.653** | **60.503** | **44.845** | **65.159** | **42.769** | **62.799** | **43.089** | **62.820** |
| Online STGCN | | 41.191 | 61.269 | 41.649 | 62.393 | 40.417 | 61.378 | 41.086 | 61.680 |
| | **+EvoMSN** | **37.592** | **55.033** | **38.236** | **56.421** | **36.626** | **54.943** | **37.485** | **55.466** |
| Online GWNET | | 38.001 | 54.350 | 38.330 | 55.565 | 39.085 | 56.332 | 38.472 | 55.416 |
| | **+EvoMSN** | **35.593** | **51.525** | **35.637** | **51.710** | **34.511** | **50.881** | **35.247** | **51.372** |

In order to evaluate the proposed method on low-frequency data, we have included **M5** competition data (Makridakis et al., 2022) for the experiment. Specifically, we conduct multivariate forecasting of aggregated daily sales of each state, each product category, each store, and each department (a total of 23 variables). We utilize the DLinear model as the forecasting backbone for both online and offline forecasting (all experiments are repeated 3 times). We compare the offline forecasting performance with/without the proposed MSN method and online forecasting performance with/without the proposed EvoMSN method in Table 7. The results show the proposed method can boost the performance on low-frequency data for both online forecasting (average reduction in MSE by **19.12%**) and offline forecasting (average reduction in MSE by **21.81%**).

Table 7: Online and Offline forecasting results on M5 dataset.

| Method | | Online DLinear | | +EvoMSN | | Offline Dlinear | | +MSN | |
|---|---|---|---|---|---|---|---|---|---|
| Metric | | MSE | MAE | MSE | MAE | MSE | MAE | MSE | MAE |
| M5 | 24 | 0.625±0.000 | 0.528±0.000 | **0.514±0.002** | **0.482±0.001** | 0.648±0.007 | 0.570±0.004 | **0.526±0.015** | **0.503±0.009** |
| | 48 | 0.658±0.000 | 0.545±0.000 | **0.537±0.001** | **0.492±0.001** | 0.717±0.003 | 0.605±0.002 | **0.542±0.009** | **0.508±0.006** |
| | 72 | 0.694±0.000 | 0.561±0.000 | **0.560±0.004** | **0.509±0.000** | 0.805±0.004 | 0.651±0.002 | **0.641±0.005** | **0.562±0.003** |
| | 96 | 0.727±0.000 | 0.575±0.000 | **0.576±0.003** | **0.515±0.002** | 0.875±0.001 | 0.679±0.001 | **0.672±0.016** | **0.574±0.010** |

To evaluate the computational efficiency of the proposed method, we have measured the computation time in seconds for each training epoch and testing batch, the time of each process in online forecasting and offline forecasting is reported in Table 8 and Table 9, respectively. The proposed method will cause extra computational complexity and running time from two perspectives. First, the proposed method requires training multi-scale statistics prediction modules, where the complexity is the number of considered periodicities times the complexity of an individual prediction module for a single periodicity. The computational complexity of this part is independent of the complexity of the forecasting backbone model and we have designed this part to be lightweight with two-layer perception networks to avoid a large computation burden. Second, the backbone forecasting model will handle the time series from multi-scale perspectives, where the complexity is the number of considered periodicities times the complexity of the backbone forecasting model. This part becomes the main computation burden when the backbone model is much more complex than the statistics prediction module. There are two possible approaches to accelerate the computation process, one is to let the multi-scale analysis in parallel (our experiment is conducted in series); another approach is to consider an attentive updation strategy in the online learning process, which is to update the model only when server distribution shift is detected instead of updating for every sample. Overall, the computational complexity of the proposed method is acceptable in many real-world applications, especially for low-frequency forecasting scenarios, such as day-ahead electricity load forecasting.

Table 8: Computation Time for Online Forecasting with EvoMSN.

| Method | | Online DLinear | | +EvoMSN | | |
|--------|----|----------------|------|------------|----------------|------|
| Time | | Train | Test | Stat Train | Backbone Train | Test |
| M5 | 24 | 0.029±0.039 | 0.002±0.000 | 0.058±0.001 | 0.083±0.001 | 0.019±0.001 |
| | 48 | 0.027±0.038 | 0.002±0.000 | 0.050±0.001 | 0.071±0.001 | 0.019±0.000 |
| | 72 | 0.026±0.039 | 0.002±0.000 | 0.048±0.002 | 0.064±0.001 | 0.020±0.003 |
| | 96 | 0.023±0.038 | 0.002±0.000 | 0.034±0.001 | 0.048±0.001 | 0.019±0.000 |

Table 9: Computation Time for Offline Forecasting with MSN.

| Method | | Offline Dlinear | | +MSN | | |
|--------|----|-----------------|------|------------|----------------|------|
| Time | | Train | Test | Stat Train | Backbone Train | Test |
| M5 | 24 | 0.086±0.039 | 0.001±0.000 | 0.247±0.003 | 0.334±0.002 | 0.004±0.000 |
| | 48 | 0.084±0.038 | 0.001±0.000 | 0.284±0.005 | 0.363±0.003 | 0.004±0.000 |
| | 72 | 0.084±0.037 | 0.001±0.000 | 0.277±0.003 | 0.357±0.005 | 0.004±0.000 |
| | 96 | 0.086±0.039 | 0.001±0.000 | 0.266±0.004 | 0.345±0.003 | 0.004±0.000 |

## A.3 FULL RESULTS OF COMPARISON OF ONLINE LEARNING STRATEGIES

This section provides comprehensive results of comparison between the proposed EvoMSN method with other online learning strategies, where the performance of forecasting horizons {96, 192, 336} are reported in Table 10. It shows the advantage of EvoMSN in enhancing online forecasting performance across all horizons. The full visualization results of online cumulative average MSE loss are shown in Fig. 7, which showcases the efficacy of EvoMSN to enable backbone models to better adapt to shifting distribution.

Table 10: Comparison between EvoMSN and existing online learning strategies. The best and second best results are in **bold** and underline.

| Methods | | EvoMSN-TCN | | Online-TCN | | FSNet | | ER | | DER++ | |
|---------|-----|------------|-------|------------|-------|-------|-------|-------|-------|-------|-------|
| Metric | | MSE | MAE | MSE | MAE | MSE | MAE | MSE | MAE | MSE | MAE |
| Exchange | 96 | **0.076** | **0.157** | 0.191 | 0.287 | 0.195 | 0.296 | 0.165 | 0.275 | 0.162 | 0.276 |
| | 192 | **0.101** | **0.178** | 0.255 | 0.325 | 0.244 | 0.327 | 0.220 | 0.310 | 0.226 | 0.318 |
| | 336 | **0.109** | **0.193** | 0.286 | 0.345 | 0.337 | 0.378 | 0.287 | 0.348 | 0.301 | 0.361 |
| Traffic | 96 | **0.480** | **0.374** | 0.851 | 0.516 | 0.941 | 0.561 | 0.838 | 0.510 | 0.842 | 0.510 |
| | 192 | **0.483** | **0.372** | 0.842 | 0.511 | 0.941 | 0.561 | 0.831 | 0.506 | 0.836 | 0.506 |
| | 336 | **0.497** | **0.377** | 0.854 | 0.514 | 0.951 | 0.565 | 0.842 | 0.509 | 0.847 | 0.509 |
| Electricity | 96 | **3.847** | **0.392** | 14.581 | 1.172 | 12.781 | 0.987 | 13.280 | 1.170 | 13.160 | 1.126 |
| | 192 | **4.166** | **0.395** | 15.279 | 1.210 | 14.486 | 1.024 | 15.190 | 1.209 | 15.234 | 1.139 |
| | 336 | **4.353** | **0.403** | 17.302 | 1.263 | 15.061 | 1.062 | 15.899 | 1.241 | 16.384 | 1.174 |
| Weather | 96 | **0.678** | **0.389** | 0.834 | 0.492 | 0.777 | 0.491 | 0.806 | 0.480 | 0.812 | 0.482 |
| | 192 | 0.737 | 0.422 | 0.774 | 0.462 | 0.913 | 0.540 | **0.697** | **0.419** | 0.709 | 0.426 |
| | 336 | 0.789 | **0.451** | 0.859 | 0.499 | 1.078 | 0.602 | **0.775** | 0.457 | 0.797 | 0.468 |
| ETTh1 | 96 | **0.600** | **0.525** | 0.748 | 0.579 | 0.614 | 0.550 | 0.741 | 0.580 | 0.684 | 0.558 |
| | 192 | 0.755 | 0.589 | 0.787 | 0.592 | **0.719** | 0.588 | 0.775 | 0.592 | 0.732 | **0.574** |
| | 336 | **0.651** | **0.552** | 0.792 | 0.594 | 0.692 | 0.575 | 0.783 | 0.593 | 0.747 | 0.578 |
| ETTm1 | 96 | **0.285** | **0.396** | 0.372 | 0.466 | 0.664 | 0.629 | 0.357 | 0.456 | 0.356 | 0.455 |
| | 192 | **0.326** | **0.423** | 0.536 | 0.553 | 0.341 | 0.444 | 0.535 | 0.552 | 0.540 | 0.555 |
| | 336 | **0.358** | **0.449** | 0.432 | 0.501 | 0.412 | 0.490 | 0.397 | 0.479 | 0.393 | 0.476 |

## A.4 FULL RESULTS OF NORMALIZATION COMPARISON IN ONLINE FORECASTING

We provide comprehensive results of comparison between the proposed EvoMSN method with other normalization strategies for online forecasting in Table 11. It can be observed that EvoMSN can achieve the best performance in most cases and its effectiveness is prominent when the forecasting horizon is long. The slice-based SAN method also shows a better performance than the instance-level normalization approaches (RevIN and DishTS), which showcases the necessity of modeling the distribution dynamics in a finer-grained manner. However, better performance can also be achieved by the instance-level normalization approaches in some cases, which shows that a coarser-grained

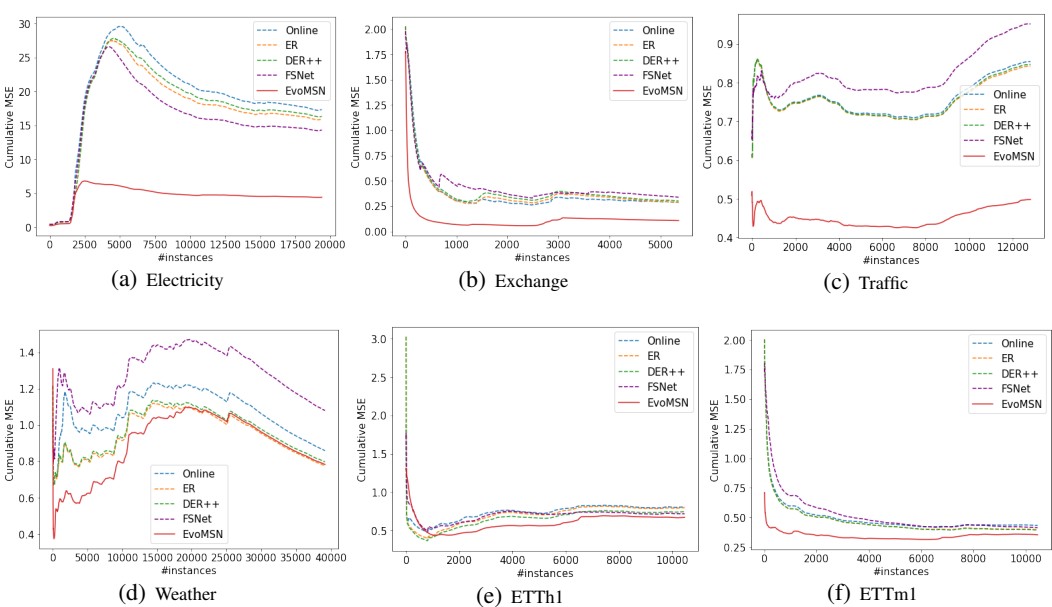

Figure 7: Evolution of the cumulative average MSE loss during online forecasting.

modeling of distribution dynamics that helps models to capture a general trend is also important. To this end, the proposed multi-scale modeling approach is a more comprehensive solution that integrates the advantages of both approaches.

Table 11: Comparison between EvoMSN and existing normalization approaches for online forecasting. The best and second best results are in **bold** and underline.

| Methods | | Autoformer | | | | | | | | TimesNet | | | | | | | |
|---|---|---|---|---|---|---|---|---|---|---|---|---|---|---|---|---|---|
| | | **+EvoMSN** | | +SAN | | +RevIN | | +Dish-TS | | **+EvoMSN** | | +SAN | | +RevIN | | +Dish-TS | |
| Metric | | MSE | MAE | MSE | MAE | MSE | MAE | MSE | MAE | MSE | MAE | MSE | MAE | MSE | MAE | MSE | MAE |
| Exchange | 96 | 0.136 | 0.223 | 0.140 | 0.217 | 0.196 | 0.202 | **0.133** | **0.200** | **0.054** | **0.143** | 0.069 | 0.143 | 0.050 | 0.129 | 0.055 | 0.150 |
| | 192 | 0.181 | 0.249 | 0.228 | 0.267 | **0.151** | **0.218** | 0.222 | 0.240 | **0.043** | **0.133** | 0.110 | 0.184 | 0.067 | 0.159 | 0.058 | 0.158 |
| | 336 | **0.227** | 0.296 | 0.289 | 0.310 | 0.233 | **0.286** | 0.339 | 0.303 | 0.091 | **0.181** | 0.163 | 0.225 | 0.096 | 0.195 | **0.080** | 0.182 |
| Traffic | 96 | **0.219** | **0.249** | 0.256 | 0.269 | 0.357 | 0.319 | 0.356 | 0.319 | **0.219** | **0.240** | 0.246 | 0.250 | 0.301 | 0.274 | 0.405 | 0.332 |
| | 192 | **0.253** | **0.256** | 0.320 | 0.295 | 0.337 | 0.300 | 0.326 | 0.298 | **0.230** | 0.240 | 0.255 | 0.258 | 0.234 | **0.235** | 0.397 | 0.325 |
| | 336 | **0.269** | **0.259** | 0.337 | 0.298 | 0.359 | 0.307 | 0.344 | 0.303 | 0.272 | 0.263 | **0.263** | **0.255** | 0.291 | 0.258 | 0.335 | 0.281 |
| Electricity | 96 | **1.358** | **0.286** | 2.211 | 0.323 | 2.782 | 0.365 | 3.123 | 0.397 | **1.684** | **0.280** | 2.221 | 0.301 | 2.038 | 0.282 | 3.420 | 0.364 |
| | 192 | **2.096** | **0.319** | 2.884 | 0.337 | 2.679 | 0.359 | 3.484 | 0.414 | **1.599** | **0.288** | 2.225 | 0.301 | 2.485 | 0.305 | 2.806 | 0.376 |
| | 336 | **2.390** | **0.321** | 3.801 | 0.362 | 4.077 | 0.416 | 4.000 | 0.421 | **2.186** | **0.312** | 2.994 | 0.336 | 3.488 | 0.343 | 3.718 | 0.412 |
| Weather | 96 | 0.973 | 0.530 | 1.026 | **0.424** | 0.984 | 0.496 | **0.920** | 0.478 | **0.407** | **0.237** | 0.648 | 0.350 | 0.699 | 0.359 | 0.748 | 0.407 |
| | 192 | **1.013** | 0.562 | 1.185 | 0.513 | 1.212 | 0.577 | 1.130 | 0.561 | **0.413** | 0.247 | 0.717 | 0.383 | 0.831 | 0.403 | 0.767 | 0.430 |
| | 336 | **1.005** | 0.566 | 1.350 | 0.570 | 1.267 | 0.606 | 1.135 | 0.579 | **0.387** | **0.237** | 0.793 | 0.421 | 1.158 | 0.441 | 0.697 | 0.422 |
| ETTh1 | 96 | **0.291** | **0.360** | 0.482 | 0.455 | 0.471 | 0.455 | 0.501 | 0.471 | **0.314** | **0.372** | 0.335 | 0.386 | 0.469 | 0.435 | 0.450 | 0.433 |
| | 192 | 0.475 | 0.457 | 0.515 | 0.474 | **0.439** | **0.444** | 0.489 | 0.472 | **0.403** | 0.424 | 0.421 | 0.432 | 0.419 | **0.412** | 0.472 | 0.453 |
| | 336 | **0.528** | 0.494 | 0.591 | 0.516 | 0.543 | **0.493** | 0.559 | 0.504 | **0.274** | **0.353** | 0.380 | 0.411 | 0.497 | 0.451 | 0.423 | 0.431 |
| ETTm1 | 96 | **0.172** | **0.300** | 0.209 | 0.332 | 0.314 | 0.413 | 0.217 | 0.342 | **0.109** | **0.236** | 0.127 | 0.258 | 0.122 | 0.250 | 0.194 | 0.324 |
| | 192 | **0.193** | **0.323** | 0.244 | 0.361 | 0.303 | 0.406 | 0.279 | 0.393 | **0.101** | **0.227** | 0.133 | 0.265 | 0.120 | 0.248 | 0.182 | 0.312 |
| | 336 | **0.267** | **0.379** | 0.287 | 0.394 | 0.327 | 0.423 | 0.276 | 0.389 | **0.102** | **0.230** | 0.153 | 0.285 | 0.114 | 0.239 | 0.131 | 0.262 |

We conduct experiments to investigate combining continual learning with different normalization methods. Specifically, we utilize FSNet (Pham et al., 2022) as the backbone and compare the vanilla FSNet, FSNet equipped with the proposed EvoMSN, with RevIN, and with DishTS in Table 12. As shown in Table 10 the proposed EvoMSN can already enable the TCN model to outperform FS-Net. Table 12 shows that the combination of EvoMSN and the continual learning strategy can further improve online forecasting performance. This is because the backbone model with a continual learning strategy can better deal with the stability-plasticity dilemma. The above results demonstrate the

compatibility of the proposed method with the continual learning strategy, and superior performance compared to other normalization methods also validates the strength of the proposed method.

Table 12: Performance comparison when different normalization methods combine with continual learning strategy.

| | | FSNet | | +EvoMSN | | +RevIN | | +Dish-TS | |
|---|---|---|---|---|---|---|---|---|---|
| | | MSE | MAE | MSE | MAE | MSE | MAE | MSE | MAE |
| | 96 | 0.195 | 0.296 | **0.065** | **0.148** | 0.322 | 0.342 | 0.459 | 0.462 |
| Exchange | 192 | 0.244 | 0.327 | **0.094** | **0.170** | 0.380 | 0.392 | 0.402 | 0.421 |
| | 336 | 0.337 | 0.378 | **0.103** | **0.193** | 0.639 | 0.518 | 1.059 | 0.686 |
| | 96 | 0.614 | 0.550 | 0.645 | **0.525** | 0.818 | 0.623 | 0.765 | 0.606 |
| ETTh1 | 192 | 0.719 | 0.588 | **0.545** | **0.494** | 0.791 | 0.618 | 0.990 | 0.689 |
| | 336 | 0.692 | 0.575 | **0.614** | **0.535** | 0.982 | 0.682 | 0.776 | 0.609 |
| | 24 | 0.664 | 0.629 | **0.326** | **0.425** | 0.674 | 0.613 | 0.635 | 0.610 |
| ETTm1 | 36 | 0.341 | 0.444 | 0.350 | **0.439** | 0.778 | 0.650 | 0.394 | 0.479 |
| | 48 | 0.412 | 0.490 | **0.350** | **0.441** | 0.509 | 0.537 | 0.452 | 0.515 |

## A.5 The Effect of Multi-Scale Normalization (MSN) for Offline Long-Term Forecasting

We investigate the traditional offline long-term forecasting setting with the train-test ratio of 70:20. This is a non-trivial task since the model will not be updated online and will suffer from the distribution discrepancy between training data and testing data. We investigate backbone models including DLinear (Zeng et al., 2023), Autoformer (Wu et al., 2021), FEDformer (Zhou et al., 2022), PatchTST (Nie et al., 2022), and TimesNet (Wu et al., 2022) on six benchmark datasets: (1) Electricity, (2) Exchange, (3) Traffic, (4) Weather, (5) Etth2, and (6) ILI [6], which collects the ratio of illness patients versus the total patients in one week that reported weekly from 2002 and 2021. We follow the same configuration as (Liu et al., 2024b) for the length of the input window and output window. In this setting, we carry out the proposed multi-scale normalization (**MSN**) method without online updation but only with offline two-stage training, where the results are reported in Table 13. By applying MSN to the backbone models, we observe an average performance improvement on all datasets across all forecasting ranges by **15.03%** with DLinear, **25.60%** with Autoformer, **22.62%** with FEDformer, **4.05%** with PatchTST, and **7.77%** with TimesNet. Such great improvements validate the effectiveness of the proposed method in explicitly modeling the distribution dynamics and removing the non-stationary factors for time series forecasting, which also shows that the proposed method can help alleviate the effects of distribution discrepancy between training and testing data to some extent. The efficacy of the proposed method is especially prominent on data that are highly non-stationary, such as Exchange, ETTh2, and ILI.

## A.6 The Comparison of MSN with Advanced Normalization Benchmarks for Offline Long-Term Forecasting

We compare the proposed MSN with other stat-of-the-art normalization methods for offline long-term forecasting, including SAN (Liu et al., 2024b), RevIN (Kim et al., 2021), and Dish-TS (Fan et al., 2023). In addition, we also include Non-Stationary Transformers (NST) (Liu et al., 2022) as a benchmark, which is designed especially for transformers to forecast non-stationary time series. We utilize FEDformer and Autoformer as the backbones and report results in Table 14. We can find that SAN performs better than the instance-level normalization methods (RevIN and Dish-TS) due to its slice-based modeling of distribution dynamics. NST shows more promising results on Weather data where other methods suffer from the over-stationarization problem. In comparison, the proposed method achieves the best performance thanks to the multi-scale modeling and ensembling.

---

[6]https://gis.cdc.gov/grasp/fluview/fluportaldashboard.html

Table 13: Offline multivariate forecasting results. The **bold** values indicate better performance when the backbone model is equipped with the proposed MSN method.

| Methods | | Dlinear | | +MSN | | Autoformer | | +MSN | | FEDformer | | +MSN | | PatchTST | | +MSN | | TimesNet | | +MSN | |
|---|---|---|---|---|---|---|---|---|---|---|---|---|---|---|---|---|---|---|---|---|---|
| Metric | | MSE | MAE | MSE | MAE | MSE | MAE | MSE | MAE | MSE | MAE | MSE | MAE | MSE | MAE | MSE | MAE | MSE | MAE | MSE | MAE |
| Exchange | 96 | 0.086 | 0.213 | **0.083** | **0.213** | 0.152 | 0.283 | **0.080** | **0.205** | 0.152 | 0.281 | **0.078** | **0.202** | 0.094 | 0.216 | **0.078** | **0.199** | 0.107 | 0.234 | **0.081** | **0.204** |
| | 192 | 0.161 | 0.297 | 0.173 | 0.317 | 0.369 | 0.437 | **0.156** | **0.297** | 0.273 | 0.380 | **0.156** | **0.298** | 0.191 | 0.311 | **0.171** | **0.300** | 0.226 | 0.344 | **0.189** | **0.320** |
| | 336 | 0.338 | 0.437 | **0.287** | **0.409** | 0.534 | 0.544 | **0.266** | **0.395** | 0.452 | 0.498 | **0.260** | **0.391** | 0.343 | 0.427 | **0.300** | **0.401** | 0.367 | 0.463 | **0.312** | **0.413** |
| | 720 | 0.999 | 0.755 | **0.662** | **0.623** | 1.222 | 0.848 | **0.604** | **0.634** | 1.151 | 0.830 | **0.608** | **0.629** | 0.888 | 0.706 | **0.832** | **0.689** | 0.964 | 0.746 | **0.914** | **0.712** |
| Traffic | 96 | 0.411 | 0.283 | 0.412 | **0.280** | 0.654 | 0.403 | **0.538** | **0.332** | 0.579 | 0.363 | **0.518** | **0.314** | 0.492 | 0.324 | **0.452** | 0.331 | 0.593 | 0.321 | **0.502** | **0.307** |
| | 192 | 0.423 | 0.289 | 0.430 | **0.286** | 0.654 | 0.410 | **0.548** | **0.338** | 0.608 | 0.376 | **0.541** | **0.326** | 0.487 | 0.303 | **0.464** | 0.325 | 0.617 | 0.336 | **0.532** | **0.324** |
| | 336 | 0.437 | 0.297 | 0.452 | 0.300 | 0.629 | 0.391 | **0.560** | **0.343** | 0.620 | 0.385 | **0.554** | **0.337** | 0.505 | 0.317 | **0.500** | 0.346 | 0.629 | 0.336 | **0.548** | **0.328** |
| | 720 | 0.467 | 0.316 | 0.484 | **0.313** | 0.657 | 0.402 | **0.605** | **0.363** | 0.630 | 0.387 | **0.579** | **0.342** | 0.542 | 0.337 | **0.540** | 0.365 | 0.640 | 0.350 | **0.574** | **0.335** |
| Electricity | 96 | 0.140 | 0.237 | **0.133** | **0.230** | 0.195 | 0.309 | **0.165** | **0.272** | 0.185 | 0.300 | **0.160** | **0.269** | 0.180 | 0.264 | **0.135** | **0.235** | 0.168 | 0.272 | **0.161** | **0.268** |
| | 192 | 0.153 | 0.250 | **0.148** | **0.245** | 0.215 | 0.325 | **0.175** | **0.282** | 0.196 | 0.310 | **0.175** | **0.284** | 0.188 | 0.275 | **0.153** | **0.255** | 0.184 | 0.289 | **0.170** | **0.277** |
| | 336 | 0.168 | 0.267 | **0.164** | **0.264** | 0.237 | 0.344 | **0.191** | **0.303** | 0.215 | 0.330 | **0.188** | **0.299** | 0.206 | 0.291 | **0.166** | **0.269** | 0.198 | 0.300 | **0.180** | **0.288** |
| | 720 | 0.203 | 0.301 | **0.200** | **0.299** | 0.292 | 0.375 | **0.226** | **0.332** | 0.244 | 0.352 | **0.206** | **0.315** | 0.247 | 0.328 | **0.203** | **0.302** | 0.220 | 0.320 | **0.201** | **0.306** |
| Weather | 96 | 0.175 | 0.237 | **0.152** | **0.214** | 0.247 | 0.320 | **0.188** | **0.255** | 0.246 | 0.328 | **0.171** | **0.240** | 0.177 | 0.218 | **0.147** | **0.206** | 0.172 | 0.220 | **0.163** | 0.224 |
| | 192 | 0.217 | 0.275 | **0.195** | **0.256** | 0.302 | 0.361 | **0.249** | **0.314** | 0.281 | 0.341 | **0.234** | **0.299** | 0.224 | 0.258 | **0.190** | **0.247** | 0.219 | 0.261 | 0.222 | 0.277 |
| | 336 | 0.263 | 0.314 | **0.245** | **0.296** | 0.362 | 0.394 | **0.317** | **0.363** | 0.337 | 0.376 | **0.302** | **0.352** | 0.277 | 0.297 | **0.243** | **0.294** | 0.280 | 0.306 | **0.273** | 0.314 |
| | 720 | 0.325 | 0.366 | **0.313** | **0.347** | 0.427 | 0.433 | **0.371** | **0.374** | 0.414 | 0.426 | **0.388** | **0.405** | 0.350 | 0.345 | **0.315** | 0.353 | 0.365 | 0.359 | 0.416 | 0.413 |
| ETTh2 | 96 | 0.292 | 0.356 | **0.276** | **0.339** | 0.384 | 0.420 | **0.308** | **0.356** | 0.341 | 0.382 | **0.297** | **0.349** | 0.294 | 0.343 | **0.284** | 0.344 | 0.340 | 0.374 | **0.296** | **0.350** |
| | 192 | 0.383 | 0.418 | **0.334** | **0.379** | 0.457 | 0.454 | **0.391** | **0.408** | 0.426 | 0.436 | **0.379** | **0.399** | 0.378 | 0.394 | **0.343** | **0.385** | 0.402 | 0.414 | **0.401** | 0.416 |
| | 336 | 0.473 | 0.477 | **0.347** | **0.397** | 0.468 | 0.473 | **0.431** | **0.444** | 0.481 | 0.479 | **0.416** | **0.433** | 0.382 | 0.410 | **0.360** | **0.403** | 0.452 | 0.452 | **0.413** | **0.435** |
| | 720 | 0.708 | 0.599 | **0.394** | **0.436** | 0.473 | 0.485 | **0.461** | **0.468** | 0.458 | 0.477 | **0.441** | **0.458** | 0.412 | 0.433 | **0.399** | 0.438 | 0.462 | 0.468 | **0.428** | **0.461** |
| ILI | 24 | 2.297 | 1.055 | **1.986** | **0.969** | 3.309 | 1.270 | **2.640** | **1.094** | 3.205 | 1.255 | **2.534** | **1.072** | 1.987 | 0.955 | **1.850** | **0.913** | 2.317 | 0.934 | **2.197** | 0.986 |
| | 36 | 2.323 | 1.070 | **1.893** | **0.898** | 3.207 | 1.216 | **2.084** | **0.904** | 3.148 | 1.288 | **2.273** | **0.942** | 1.872 | 0.893 | **1.827** | **0.873** | 1.972 | 0.920 | **1.686** | 0.849 |
| | 48 | 2.262 | 1.065 | **1.895** | **0.925** | 3.166 | 1.198 | **2.182** | **0.943** | 2.913 | 1.168 | **2.262** | **0.961** | 1.840 | 0.900 | 1.843 | **0.885** | 2.238 | 0.940 | **1.973** | **0.885** |
| | 60 | 2.443 | 1.124 | **2.062** | **0.980** | 2.947 | 1.159 | **2.307** | **0.982** | 2.853 | 1.161 | **2.069** | **0.917** | 2.021 | 0.961 | 2.192 | 0.988 | 2.027 | 0.928 | 2.072 | 0.940 |

Table 14: Comparison between MSN and existing normalization approaches for offline forecasting. The best and second best results are in **bold** and underline.

| Methods | | FEDformer | | | | | | | | | | Autoformer | | | | | | | | | |
|---|---|---|---|---|---|---|---|---|---|---|---|---|---|---|---|---|---|---|---|---|---|
| | | +MSN | | +SAN | | +RevIN | | +NST | | +Dish-TS | | +MSN | | +SAN | | +RevIN | | +NST | | +Dish-TS | |
| Metric | | MSE | MAE | MSE | MAE | MSE | MAE | MSE | MAE | MSE | MAE | MSE | MAE | MSE | MAE | MSE | MAE | MSE | MAE | MSE | MAE |
| Exchange | 96 | **0.078** | **0.202** | 0.079 | 0.205 | 0.148 | 0.279 | 0.145 | 0.275 | 0.131 | 0.263 | **0.080** | **0.205** | 0.082 | 0.208 | 0.166 | 0.295 | 0.177 | 0.304 | 0.225 | 0.341 |
| | 192 | **0.156** | 0.298 | **0.156** | **0.295** | 0.266 | 0.377 | 0.274 | 0.383 | 0.538 | 0.523 | **0.156** | 0.297 | 0.157 | **0.296** | 0.299 | 0.404 | 0.275 | 0.385 | 0.760 | 0.610 |
| | 336 | **0.260** | **0.391** | **0.260** | 0.384 | 0.428 | 0.484 | 0.437 | 0.488 | 0.667 | 0.591 | 0.266 | **0.395** | **0.262** | 0.385 | 0.448 | 0.496 | 0.442 | 0.490 | 0.707 | 0.628 |
| | 720 | **0.608** | **0.629** | 0.697 | 0.633 | 1.056 | 0.789 | 1.064 | 0.787 | 1.480 | 0.954 | **0.604** | **0.634** | 0.689 | 0.629 | 1.068 | 0.791 | 1.049 | 0.784 | 2.341 | 1.063 |
| Traffic | 96 | **0.518** | **0.314** | 0.536 | 0.330 | 0.613 | 0.347 | 0.612 | 0.348 | 0.613 | 0.350 | **0.538** | **0.332** | 0.569 | 0.350 | 0.643 | 0.354 | 0.645 | 0.354 | 0.652 | 0.363 |
| | 192 | **0.541** | **0.326** | 0.565 | 0.345 | 0.637 | 0.356 | 0.641 | 0.357 | 0.644 | 0.362 | **0.548** | **0.338** | 0.594 | 0.364 | 0.659 | 0.373 | 0.643 | 0.367 | 0.669 | 0.374 |
| | 336 | **0.554** | **0.337** | 0.580 | 0.354 | 0.652 | 0.363 | 0.654 | 0.363 | 0.659 | 0.370 | **0.560** | **0.343** | 0.591 | 0.363 | 0.662 | 0.371 | 0.665 | 0.363 | 0.683 | 0.376 |
| | 720 | **0.579** | **0.342** | 0.607 | 0.367 | 0.686 | 0.382 | 0.688 | 0.380 | 0.693 | 0.388 | **0.605** | **0.363** | 0.623 | 0.380 | 0.700 | 0.384 | 0.667 | 0.373 | 0.703 | 0.392 |
| Electricity | 96 | **0.160** | **0.269** | 0.164 | 0.272 | 0.172 | 0.278 | 0.172 | 0.279 | 0.175 | 0.284 | **0.165** | **0.272** | 0.172 | 0.281 | 0.179 | 0.286 | 0.179 | 0.285 | 0.179 | 0.290 |
| | 192 | **0.175** | **0.284** | 0.179 | 0.286 | 0.185 | 0.289 | 0.187 | 0.291 | 0.188 | 0.296 | **0.175** | **0.282** | 0.195 | 0.300 | 0.216 | 0.316 | 0.209 | 0.309 | 0.215 | 0.318 |
| | 336 | **0.188** | **0.299** | 0.191 | 0.299 | 0.200 | 0.304 | 0.202 | 0.307 | 0.209 | 0.319 | **0.191** | **0.303** | 0.211 | 0.316 | 0.233 | 0.331 | 0.246 | 0.335 | 0.244 | 0.343 |
| | 720 | **0.206** | **0.315** | 0.230 | 0.334 | 0.243 | 0.337 | 0.230 | 0.326 | 0.239 | 0.343 | **0.226** | **0.332** | 0.236 | 0.335 | 0.246 | 0.341 | 0.252 | 0.345 | 0.286 | 0.370 |
| Weather | 96 | **0.171** | 0.240 | 0.179 | 0.239 | 0.187 | **0.234** | 0.187 | **0.234** | 0.244 | 0.317 | **0.188** | 0.255 | 0.194 | 0.256 | 0.212 | 0.257 | 0.211 | **0.254** | 0.268 | 0.338 |
| | 192 | **0.234** | 0.299 | 0.234 | 0.296 | 0.235 | **0.272** | 0.235 | **0.272** | 0.320 | 0.380 | **0.249** | 0.314 | 0.258 | 0.316 | 0.264 | 0.300 | 0.265 | **0.301** | 0.376 | 0.421 |
| | 336 | 0.302 | 0.352 | 0.304 | 0.348 | **0.287** | **0.307** | 0.289 | 0.308 | 0.424 | 0.452 | 0.317 | 0.363 | 0.329 | 0.367 | 0.309 | 0.329 | **0.303** | **0.324** | 0.486 | 0.486 |
| | 720 | 0.388 | 0.405 | 0.400 | 0.404 | 0.361 | 0.353 | 0.359 | **0.352** | 0.604 | 0.553 | **0.371** | 0.374 | 0.440 | 0.438 | 0.377 | 0.367 | 0.366 | **0.357** | 0.612 | 0.560 |
| ETTh2 | 96 | **0.297** | **0.349** | 0.300 | 0.355 | 0.380 | 0.402 | 0.381 | 0.403 | 0.806 | 0.589 | **0.308** | **0.356** | 0.316 | 0.366 | 0.411 | 0.410 | 0.394 | 0.398 | 1.100 | 0.670 |
| | 192 | **0.379** | **0.399** | 0.392 | 0.413 | 0.457 | 0.443 | 0.478 | 0.453 | 0.936 | 0.659 | **0.391** | **0.408** | 0.413 | 0.426 | 0.478 | 0.450 | 0.473 | 0.450 | 0.976 | 0.672 |
| | 336 | **0.416** | **0.433** | 0.459 | 0.462 | 0.515 | 0.479 | 0.561 | 0.499 | 1.039 | 0.702 | **0.431** | **0.444** | 0.446 | 0.457 | 0.545 | 0.493 | 0.528 | 0.490 | 1.521 | 0.783 |
| | 720 | **0.441** | **0.458** | 0.462 | 0.472 | 0.507 | 0.487 | 0.502 | 0.487 | 1.237 | 0.759 | **0.461** | **0.468** | 0.471 | 0.474 | 0.523 | 0.490 | 0.524 | 0.490 | 1.105 | 0.745 |
| ILI | 24 | **2.534** | **1.072** | 2.614 | 1.119 | 3.218 | 1.172 | 3.302 | 1.281 | 2.883 | 1.102 | **2.640** | **1.094** | 2.777 | 1.157 | 3.780 | 1.270 | 3.482 | 1.207 | 3.636 | 1.249 |
| | 36 | **2.273** | **0.942** | 2.537 | 1.079 | 3.055 | 1.135 | 3.193 | 1.240 | 2.865 | 1.077 | **2.084** | **0.904** | 2.649 | 1.104 | 3.114 | 1.157 | 3.423 | 1.289 | 3.284 | 1.178 |
| | 48 | **2.262** | **0.961** | 2.416 | 1.032 | 2.734 | 1.055 | 2.936 | 1.171 | 2.759 | 1.033 | **2.182** | **0.943** | 2.420 | 1.029 | 2.865 | 1.099 | 3.163 | 1.217 | 2.942 | 1.086 |
| | 60 | **2.069** | **0.917** | 2.299 | 1.003 | 2.841 | 1.095 | 2.904 | 1.173 | 2.878 | 1.075 | **2.307** | **0.982** | 2.401 | 1.021 | 2.846 | 1.104 | 2.871 | 1.140 | 2.856 | 1.083 |

## B LIMITATION

Even though the proposed method can greatly improve the overall forecasting accuracy, it may sometimes cause slight performance degradation. We conclude potential reasons for method failure as follows: 1) In the online forecasting setting, the multi-scale statistics prediction module is updated only once for each incoming new data, which may challenge the module in capturing some fast-changing statistics and will result in bad performance. Besides, the global dominant periodicity is determined according to the training data, where the relatively small training data split ratio in the online setting may cause an inappropriate periodicity extraction and thus affect the effectiveness of multi-scale slice-based analysis. 2) In the offline forecasting setting, the multi-scale statistics pre-

diction module is only updated on the training data, where the distinct statistics evolving dynamics of training data and testing data may cause a worse performance in the evaluation stage. Despite the potential shortcomings identified above, it is essential to emphasize the overall strengths of the proposed method.

Moreover, we point out two directions to further improve the proposed method. First, we only model the distribution dynamics by investigating slice statistics including mean and deviation, where more comprehensive characteristics of distribution, such as minimum and maximum value, are important but neglected in the proposed framework. We will propose a more general approach to model data distribution dynamics in our future work. Second, the proposed EvoMSN may suffer from the stability-plasticity dilemma when the backbone model and statistics prediction module are updated online, where incremental learning techniques could be further investigated and integrated into the proposed framework to improve the online forecasting performance.

