**Table 1: Extended benchmarks for online forecasting performance comparison with/without EvoMSN.**

| Methods | | TimeMixer | | +EvoMSN | | PatchMixer | | +EvoMSN | | Koopa | | +EvoMSN | | iTransformer | | +EvoMSN | | TSMixer | | +EvoMSN | |
|---|---|---|---|---|---|---|---|---|---|---|---|---|---|---|---|---|---|---|---|---|---|
| Metric | | MSE | MAE | MSE | MAE | MSE | MAE | MSE | MAE | MSE | MAE | MSE | MAE | MSE | MAE | MSE | MAE | MSE | MAE | MSE | MAE |
| Exchange | 96 | 0.132 | 0.236 | **0.105** | **0.218** | 0.120 | 0.222 | **0.090** | **0.181** | 0.076 | 0.172 | **0.054** | **0.158** | 0.124 | 0.223 | **0.102** | **0.212** | 0.103 | 0.214 | **0.069** | **0.175** |
| | 192 | 0.235 | 0.319 | **0.180** | **0.278** | 0.232 | 0.308 | **0.124** | **0.213** | 0.101 | 0.200 | **0.064** | **0.171** | 0.229 | 0.304 | **0.159** | **0.265** | 0.131 | 0.245 | **0.081** | **0.190** |
| | 336 | 0.353 | 0.400 | **0.243** | **0.322** | 0.324 | 0.375 | **0.158** | **0.237** | 0.132 | 0.232 | **0.078** | **0.188** | 0.353 | 0.388 | **0.220** | **0.305** | 0.133 | 0.251 | **0.101** | **0.210** |
| ETTm1 | 96 | 0.326 | 0.426 | **0.307** | **0.418** | 0.310 | 0.413 | **0.290** | **0.400** | 0.249 | 0.367 | **0.247** | 0.372 | 0.301 | 0.402 | **0.293** | **0.405** | 0.207 | 0.339 | **0.203** | **0.331** |
| | 192 | 0.386 | 0.461 | **0.322** | **0.424** | 0.293 | 0.393 | **0.281** | **0.386** | 0.259 | 0.376 | **0.257** | 0.379 | 0.337 | 0.423 | **0.315** | **0.415** | 0.261 | 0.388 | **0.219** | **0.350** |
| | 336 | 0.467 | 0.407 | **0.325** | **0.428** | 0.295 | 0.388 | **0.230** | **0.348** | 0.288 | 0.399 | **0.261** | **0.384** | 0.422 | 0.469 | **0.347** | **0.436** | 0.309 | 0.425 | **0.215** | **0.348** |

**Table 2: Extended sensitivity analysis on the number of multi-scales $K$ with TSMixer as the backbone.**

| Number of Scales | | $K=1$ | | $K=2$ | | $K=3$ | | $K=4$ | | $K=6$ | | $K=8$ | |
|---|---|---|---|---|---|---|---|---|---|---|---|---|---|
| Metric | | MSE | MAE | MSE | MAE | MSE | MAE | MSE | MAE | MSE | MAE | MSE | MAE |
| Exchange | 96 | 0.123 | 0.204 | 0.118 | 0.218 | 0.085 | 0.192 | 0.069 | 0.175 | 0.056 | 0.159 | 0.060 | 0.160 |
| | 192 | 0.181 | 0.284 | 0.119 | 0.229 | 0.086 | 0.203 | 0.081 | 0.190 | 0.082 | 0.180 | 0.065 | 0.169 |
| | 336 | 0.350 | 0.380 | 0.144 | 0.254 | 0.105 | 0.224 | 0.101 | 0.210 | 0.088 | 0.190 | 0.080 | 0.183 |
| ETTm1 | 96 | 0.262 | 0.376 | 0.245 | 0.369 | 0.213 | 0.343 | 0.203 | 0.331 | 0.198 | 0.318 | 0.171 | 0.304 |
| | 192 | 0.232 | 0.360 | 0.231 | 0.358 | 0.238 | 0.365 | 0.219 | 0.350 | 0.196 | 0.329 | 0.175 | 0.308 |
| | 336 | 0.254 | 0.376 | 0.253 | 0.377 | 0.268 | 0.389 | 0.215 | 0.348 | 0.215 | 0.342 | 0.190 | 0.323 |

**Table 3: Extended normalization comparison for online forecasting.**

| Methods | | TSMixer | | | | | | | | PatchMixer | | | | | | | |
|---|---|---|---|---|---|---|---|---|---|---|---|---|---|---|---|---|---|
| | | +EvoMSN | | +SAN | | +RevIN | | +Dish-TS | | +EvoMSN | | +SAN | | +RevIN | | +Dish-TS | |
| Metric | | MSE | MAE | MSE | MAE | MSE | MAE | MSE | MAE | MSE | MAE | MSE | MAE | MSE | MAE | MSE | MAE |
| Exchange | 96 | **0.069** | **0.175** | 0.076 | 0.175 | 0.080 | 0.185 | 0.091 | 0.203 | **0.090** | **0.181** | 0.101 | 0.213 | 0.120 | 0.222 | 0.187 | 0.283 |
| | 192 | **0.081** | **0.190** | 0.099 | 0.200 | 0.096 | 0.206 | 0.120 | 0.238 | **0.124** | **0.213** | 0.168 | 0.275 | 0.232 | 0.308 | 0.293 | 0.351 |
| | 336 | **0.101** | **0.210** | 0.113 | 0.217 | 0.109 | 0.223 | 0.149 | 0.266 | **0.158** | **0.237** | 0.253 | 0.333 | 0.324 | 0.375 | 0.323 | 0.370 |
| ETTm1 | 96 | **0.203** | **0.331** | 0.262 | 0.373 | 0.290 | 0.408 | 0.274 | 0.395 | **0.290** | **0.400** | 0.351 | 0.426 | 0.310 | 0.413 | 0.402 | 0.479 |
| | 192 | **0.219** | **0.350** | 0.276 | 0.386 | 0.303 | 0.420 | 0.300 | 0.419 | **0.281** | **0.386** | 0.305 | 0.401 | 0.293 | 0.393 | 0.390 | 0.473 |
| | 336 | **0.215** | **0.348** | 0.344 | 0.428 | 0.330 | 0.444 | 0.339 | 0.441 | **0.230** | **0.348** | 0.355 | 0.428 | 0.295 | 0.388 | 0.415 | 0.487 |

**Table 4: Extended online forecasting results on largest traffic dataset (2017-2021).**

| | Pred_len 48 | | Pred_len 72 | | Pred_len 96 | | Avg | |
|---|---|---|---|---|---|---|---|---|
| | MAE | RMSE | MAE | RMSE | MAE | RMSE | MAE | RMSE |
| Online LSTM | 35.014 | 52.289 | 35.445 | 53.029 | 36.868 | 54.645 | 35.776 | 53.321 |
| **+EvoMSN** | **29.652** | **43.668** | **30.104** | **44.854** | **29.317** | **44.053** | **29.691** | **44.191** |
| *Improvement* | 15.32% | 16.49% | 15.07% | 15.42% | 20.48% | 19.38% | 17.01% | 17.12% |

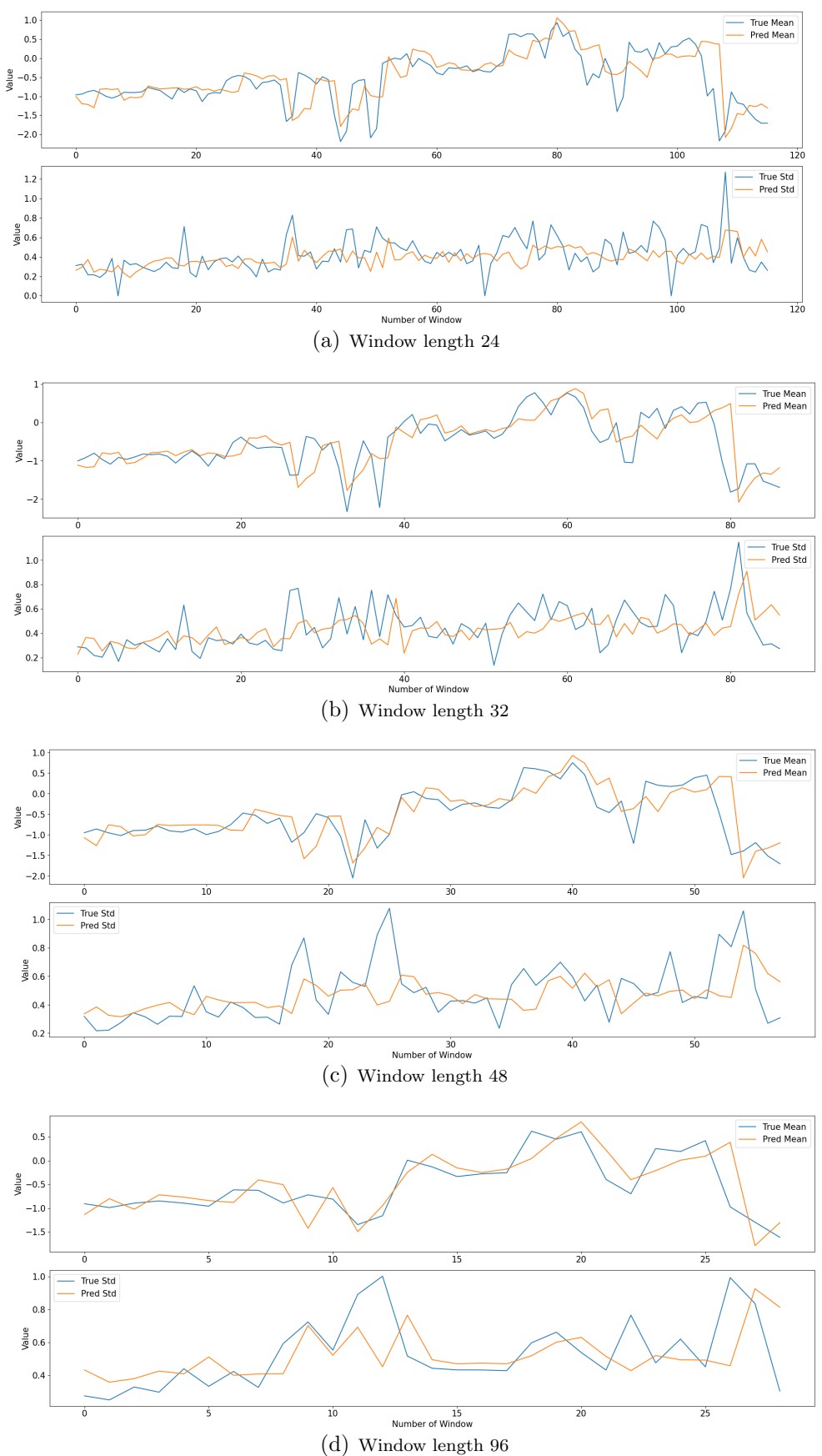

Figure 1: Visualization of predicted statistics. (a), (b), (c), (d) plot the predicted and true statistics of each window when the window length equals 24, 32, 48, 96, respectively.