# OpenReview forum: "Evolving Multi-Scale Normalization for Time Series Forecasting under Distribution Shifts"
_ICLR.cc/2025/Conference — ICLR 2025 Conference Withdrawn Submission_

### Official Review · Reviewer_xqUB · 2024-10-28

**Soundness:** 2
**Presentation:** 3
**Contribution:** 2
**Rating:** 3
**Confidence:** 4

**Summary:**

The author proposed a new model-agnostic Evolutionary Multi-Scale Normalization (EvoMSN) framework to address the distribution shift issue. It includes a multi-scale statistical prediction module and flexible normalization and denormalization methods with adaptive integration. The experiments demonstrated the effectiveness of the model.

**Strengths:**

Time series prediction is very important.

The author proposes an effective model.

**Weaknesses:**

Q1. The relevant experimental details are confusing. What does the statement 'According to the widely applied online setting, the forecasts will be made as each test data sample arrives, and the model will be updated by one epoch according to the online forecasting loss' mean? Do you update the model once for a test sample or once for a batch sample? Please further explain the specific settings of online learning in detail.

Q2.  To validate the model's effectiveness across a broader spectrum, larger-scale datasets spanning longer time periods should be incorporated.  This will help ascertain the model's robustness and applicability in scenarios with more significant temporal variations.

Q3. Could authors add more baselines in the experiment, such as PatchMixer, Crossgcn, iTransformer, TimeMixer, Koopa, and other models.

Q4. The author's contribution is incremental. The author's online learning strategy mainly involves patch normalization and two-stage training strategies, which have already been discussed in SAN and [2]. The contributions of SAN and [2] were overlooked in the related work and methods section. Additionally, what are the differences between the evolutionary training strategy and [2]? In Table 14, I did not observe significant performance improvements of the proposed model compared to SAN. The idea of multi-scale modeling originates from TimeMixer. It is recommended that the author further discuss these works in detail to clearly demonstrate their contribution. Furthermore, I suggest rewriting the recommendation section to highlight the technical gaps in existing works more clearly to clarify the motivation.

Q5. In Table 5, we can see that the primary performance of the proposed method comes from the Online-strategy based on BI-LEVEL OPTIMIZATION, as discussed in [1] and [2]. Additionally, why did the author not evaluate the performance variability of multi-scale modeling variables? What findings are observed when K is greater than 4? Why were only ablation experiments conducted on DLinear? As far as I know, DLinear is a small model with unstable training, where even a small difference in a epoch could lead to significant performance variations. Therefore, it is recommended to add more benchmarks for a comprehensive evaluation.

Q5. Time series datasets are typically offline, and the running time of various models does not exceed one day.  Could you elaborate on the practical applications and advantages of online time series learning?

Q6. DLinear reported a performance of 0.081 in the Exchange but 0.131 in this paper, and the performance of other benchmarks also dropped significantly in this paper. What could be the reason for this? A significant decline in baseline performance also occurs in Table 13. Please explain potential reasons for the discrepancy in results.

Q7. Can the author provide a comparison of other models in Table 11?

Q8. The author's motivation for the paper is the non-stationarity of time series, and the contribution is to address this problem from an online learning perspective. However, by comparing Table 1 and Table 8, I find that online learning does not actually have any benefits, and the performance of most models has shown a huge decline. The potential reason is that a test sample will lead to overfitting after updating the model. We do not need this update, and the original model can perform very well (at least better than after online learning).


Q9. Unfair comparison. As mentioned above, updating the model with data from one batch or one sample can lead to overfitting. The baseline model was updated twice, while the author's model was actually updated only once, resulting in a delayed decrease in model performance.

Ref:
[1] Adaptive Normalization for Non-stationary Time Series Forecasting: A Temporal Slice Perspective.
[2] Stephen Gould, Basura Fernando, Anoop Cherian, Peter Anderson, Rodrigo Santa Cruz, and Edison Guo. 2016. On differentiating parameterized argmin and argmax problems with application to bi-level optimization. arXiv preprint arXiv:1607.05447 (2016).

**Questions:**

See Weaknesses.

---

> ### Author Response · Authors · 2024-11-21
>
> We are grateful for your thorough examination of our manuscript and the opportunity to clarify and enhance our work based on your insightful comments. We would like to address your concerns as follows.
>
> - Q1:
>
> Sorry for the unclear description. In the online setting, we set the test_batchsize=1, thus the model is updated for each test sample. We have made it clearer in the manuscript.
>
> - Q2:
>
> **We have already demonstrated the effectiveness of the proposed method on the largest traffic dataset [1] in Table 6** (where online forecasting in the year 2019 is conducted). We would like to strengthen this statement further by reconducting the experiment on this dataset and **spanning a longer period (5 years, from 2017 to 2021)**, where results again verify the robustness of the proposed method.
>
> |||Online_LSTM||
> |-|-|-|-|
> |||w/o EvoMSN|**+EvoMSN**|
> |Pred_len 48|MAE|35.014|**29.652**|
> ||RMSE|52.289|**43.668**|
> |Pred_len 72|MAE|35.445|**30.104**|
> ||RMSE|53.029|**44.854**|
> |Pred_len 96|MAE|36.868|**29.317**|
> ||RMSE|54.645|**44.053**|
> |Avg|MAE|35.776|**29.691**|
> ||RMSE|53.321|**44.191**|
>
> [1] Largest: A benchmark dataset for large-scale traffic forecasting
>
> - Q3:
>
> We agree that comparing with a wider range of baselines would strengthen our experimental results. We apply the proposed method to backbone models including PatchMixer, TimeMixer, iTransformer, Koopa, and TSMixer (as the Crossgcn is not designed for time series forecasting, we adopt TSMixer instead) for **online forecasting**. The results are shown in the following table (The average performance for different prediction lengths \{96,192,336\} is reported). The results demonstrate the broad effectiveness of the proposed methods, detailed results are reported in Table 1 in the uploaded **supplementary material**.
>
> |||Exchange||ETTm||
> |-|-|-|-|-|-|
> |||MSE|MAE|MSE|MAE|
> |TimeMixer||0.240|0.318|0.393|0.431|
> ||**+EvoMSN**|**0.176**|**0.273**|**0.318**|**0.423**|
> |PatchMixer||0.225|0.302|0.299|0.398|
> ||**+EvoMSN**|**0.124**|**0.210**|**0.267**|**0.378**|
> |Koopa||0.103|0.201|0.266|0.381|
> ||**+EvoMSN**|**0.065**|**0.172**|**0.255**|**0.378**|
> |iTransformer||0.235| 0.305 | 0.353 | 0.431  |
> ||**+EvoMSN**|**0.160**|**0.260**|**0.318**|**0.419**|
> |TSMixer||0.122|0.237|0.259|0.384|
> ||**+EvoMSN**|**0.084**|**0.192**|**0.212**|**0.343**|
>
> - Q4:
>
> We believe integrating **multiscale information for normalization and online learning** represents a key contribution. **Even though SAN adopts a similar slice-based normalization from work, it relies on single-scale modeling and also neglects the distribution shift problem over time.** Our method enhances it by multi-scale modeling with adaptive ensembling and alternative online learning strategy, where both parts are validated to be superior to SAN. Taking Autoformer as the backbone forecasting model, **for offline forecasting**, Table 14 shows the EvoMSN achieves an average improvement (overall all datasets and prediction lengths) of **6.23\%** in terms of MSE compared to SAN, which demonstrates the strength of the proposed multiscale approach. **For online forecasting**, Table 11 shows the EvoMSN achieves a more prominent average improvement of **26.33\%** in terms of MSE compared to SAN, which indicates modeling distribution dynamics with a multiscale view is superior and more robust than a single view approach that adopted by SAN for addressing the distribution shifts problem in a data stream. We reckon these improvements compared to SAN  are non-trivial.
>
> Even though there are some existing studies investigating multi-scale modeling, such as TimeMixer and TimesNet, our design focuses on a totally different emphasis. Compared to these methods, our proposed multi-scale modeling does not directly let the backbone forecasting model handle the raw time series. Instead, **we depend on the multi-periodicity information to explicitly model the dynamics of data distribution (statistics information includes mean and standard deviations)**, which is the first study to do so to the best of our knowledge. Different from existing studies that design specific model architecture, our design enhances the performance of arbitrary backbone forecasting models, and the effectiveness is also shown when it is applied to TimeMixer and TimesNet.
>
> We have carefully read the recommended reference [1] and we find the only relation to our study is the form of bi-level optimization formulation. However, their focus is on general bi-level optimization instead of specific time series forecasting problems, and the methodology has nothing in common with that of ours.
>
> [1] On differentiating parameterized argmin and argmax problems with application to bi-level optimization.

---

> > ### Author Response · Authors · 2024-11-21
> >
> > - Q5:
> >
> > We would like to clarify that your mentioned references haven't analyzed or applied the bi-level optimization problem to an online setting, which differentiates our study from these works. To evaluate the performance of the multi-scale modeling approach, we have compared it to the approach without multi-scale modeling (Table 1) and to the approach with single-scale modeling (Table 4). We appreciate your suggestion to use the advanced model instead of DLinear to conduct sensitivity analysis on the number of scales $K$. To this end, we use TSMixer as the backbone model and evaluate the performance of EvoMSN with different $K=\{1, 2, 3, 4, 6, 8\}$ in the following table. We again find the overall trend that the model performance improves when more scales are considered to model the distribution dynamics. When the $K$ is too small, the statistics prediction module may not robustly capture the complex distribution dynamics. But a too-large $K$ may also be problematic because the computational burden will increase linearly according to $K$ and a large $K$ will result in short slices that may cause the estimation of statistics inaccurate.
> >
> > |Number of Scales||K=1||K=2||K=3||K=4||K=6||K=8||
> > |-|-|-|-|-|-|-|-|-|-|-|-|-|-|
> > |Metric||MSE|MAE|MSE|MAE|MSE|MAE|MSE|MAE|MSE|MAE|MSE|MAE|
> > |Exchange|96|0.123|0.204|0.118|0.218|0.085|0.192|0.069|0.175|0.056|0.159|0.060|0.160|
> > ||192|0.181|0.284|0.119|0.229|0.086|0.203|0.081|0.190|0.082|0.180|0.065|0.169|
> > ||336|0.350|0.380|0.144|0.254|0.105|0.224|0.101|0.210|0.088|0.190|0.080|0.183|
> > |ETTm1|96|0.262|0.376|0.245|0.369|0.213|0.343|0.203|0.331|0.198|0.318|0.171|0.304|
> > ||192|0.232|0.360|0.231|0.358|0.238|0.365|0.219|0.350|0.196|0.329|0.175|0.308|
> > ||336|0.254|0.376|0.253|0.377|0.268|0.389|0.215|0.348|0.215|0.342|0.190|0.323|
> >
> > - Q5:
> >
> > Online time series learning has practical applications and advantages that are particularly relevant in scenarios where data is generated and arrives sequentially over time, and immediate decisions or predictions are required, such as stock market analysis, weather forecasting, and network traffic management. Unlike offline learning, which processes all data at once, online learning is designed to handle data streams. This is particularly useful for large-scale applications where data is not only vast but also arrives at a high velocity, making traditional batch-learning methods inefficient and impractical. Besides, Online learning models can be updated incrementally as new data arrives, without the need to retrain the model from scratch.
> >
> > - Q6:
> >
> > We would like to kindly point out that you have compared the online forecasting performance to the offline forecasting performance reported in existing studies, which is incomparable and seems like a 'performance discrepancy'. Actually, **for offline forecasting benchmarks, we refer to the performances reported in [1] (Table 3)**, where DLinear performance is reported as 0.086. Other offline benchmark performances are also consistent with the reported ones, please compare Table 13 in our paper with Table 3 in [1].  For online forecasting benchmarks, we run the experiment based on models with their default configurations, where the DLinear has a performance of 0.131 on the exchange dataset.
> >
> > [1] Adaptive Normalization for Non-stationary Time Series Forecasting: A Temporal Slice Perspective.
> >
> > - Q7:
> >
> > We have included TSMixer and PatchMixer as backbone models to compare different normalization approaches for online forecasting in the following tables. Results again demonstrate the superiority of the proposed EvoMSN.
> >
> > |TSMixer||**+EvoMSN**||+SAN||+RevIN||+Dish-TS ||
> > |-|-|-|-|-|-|-|-|-|-|
> > |||MSE|MAE|MSE|MAE|MSE|MAE|MSE|MAE|
> > |Exchange|96|**0.069**|**0.175**|0.076|0.175|0.080|0.185|0.091|0.203|
> > ||192|**0.081**|**0.190**|0.099|0.200|0.096|0.206|0.120|0.238|
> > ||336|**0.101**|**0.210**|0.113|0.217|0.109|0.223|0.149|0.266|
> > |ETTm1|96|**0.203**|**0.331**|0.262|0.373|0.290|0.408|0.274|0.395|
> > ||192|**0.219**|**0.350**|0.276|0.386|0.303|0.420|0.300|0.419|
> > ||336|**0.215**|**0.348**|0.344|0.428|0.330|0.444|0.339|0.441|
> >
> > | PatchMixer||**+EvoMSN**||+SAN||+RevIN||+Dish-TS||
> > |-|-|-|-|-|-|-|-|-|-|
> > |||MSE|MAE|MSE|MAE|MSE|MAE|MSE|MAE|
> > |Exchange|96|**0.090**|**0.181**|0.101|0.213|0.120|0.222|0.187|0.283|
> > ||192|**0.124**|**0.213**|0.168|0.275|0.232|0.308|0.293|0.351|
> > ||336|**0.158**|**0.237**|0.253|0.333|0.324|0.375|0.323|0.370|
> > |ETTm1|96|**0.290**|**0.400**|0.351|0.426|0.310|0.413|0.402|0.479|
> > ||192|**0.281**|**0.386**|0.305|0.401|0.293|0.393|0.390|0.473|
> > ||336|**0.230**|**0.348**|0.355|0.428|0.295|0.388|0.415|0.487|

---

> > > ### Author Response · Authors · 2024-11-21
> > >
> > > - Q8 & Q9:
> > >
> > > The contribution of this study to address the distribution shifts problem is to propose multiscale normalization for online learning. **For Table 1**, results (in terms of MSE) show **77/90** cases have performance improvement with an average improvement of **15.99\%** for Autoformer, **23.24\%** for FEDformer, **12.78\%** for DLinear, **9.23\%** for PatchTST, **14.23\%** for TimesNet. We believe these improvements show the benefits of the proposed methods. **For Table 8**, it shows the computational time of the proposed methods instead of forecasting performance.
> > >
> > > We respectfully disagree with your assertion that online learning is unnecessary. We have compared forecasting performance with or without online updation in Table 5 (the first column is the offline approach without online update and the second column is the approach to online update backbone model), where results show the online update achieves an average improvement of **10.18\%** in terms of MSE compared to the offline one, showing the necessity of online learning. To this end, your concern about unfair comparison doesn't exist as online learning is beneficial to address the distribution shifts problem and we have carefully experimented to ensure a fair and accurate comparison.

---

> > > > ### Comment · Reviewer_xqUB · 2024-11-23
> > > > **Thank you**
> > > >
> > > > Thank you for your detailed response. I still have the following concerns:
> > > >
> > > > (1) Novelty. The authors apply slicing to continuous learning, but slicing techniques have been widely applied in many models - for example, I don't see significant differences from SAN.
> > > >
> > > > (2) Q2. Why did the authors only use LSTM? What about other backbones? Can the authors report results for 3, 6, and 12 time granularities? What is the dataset setup?
> > > >
> > > > (3) Q5. Can you directly tell me where online learning is implemented? What are the advantages of online learning to your knowledge? From what I understand, time prediction tasks are not time-sensitive, so the authors' subjective decision to make dataset divisions very small is fatal for most models because they need sufficient data to fit, naturally leading to poor performance. Why can't we collect a certain amount of data before training the model? Additionally, isn't the rule of one test and one epoch of online training too strict? For traffic data, one sampling point is only five minutes, and training one epoch takes just over ten seconds. This makes me seriously doubt the practical application of the authors' research - do we really need to execute it this way?
> > > >
> > > > (4) The authors' writing is disconnected. They attempt to address temporal shift using online learning strategies, but it's only effective in their customized settings. Perhaps I suggest the authors consider the problem from the perspective of temporal online learning, summarizing the necessity and gaps in online learning.
> > > >
> > > > (5) The authors haven't addressed my concerns about fairness in comparisons. In fact, the authors' algorithm essentially updates every two epochs, which reduces the rate of performance degradation.

---

### Official Review · Reviewer_KoUF · 2024-10-28

**Soundness:** 4
**Presentation:** 3
**Contribution:** 3
**Rating:** 5
**Confidence:** 3

**Summary:**

The paper presents the EvoMSN framework, a novel approach to time series forecasting under distribution shifts. It introduces a model-agnostic methodology that utilizes multi-scale statistics prediction and adaptive ensembling to normalize and denormalize data, addressing the complexities of distribution dynamics and non-stationarity. The framework is evaluated through comprehensive experiments on real-world datasets, demonstrating significant improvements over the original models.

**Strengths:**

1. The proposal to use multi-scale statistics prediction and adaptive ensembling for normalization is innovative and addresses a critical gap in handling distribution shifts in time series data. And the framework is model-agnostic, increasing its versatility and applicability across various forecasting models.
2. The paper provides a thorough experimental evaluation, showcasing the effectiveness of EvoMSN in improving forecasting performance under distribution shifts across different datasets and forecasting models as well as other frameworks. The results demonstrate substantial improvements over state-of-the-art methods.

**Weaknesses:**

1. The paper could benefit from a more detailed discussion on the limitations of the EvoMSN framework, such as assumptions made, potential scenarios where the framework may underperform, and its scalability challenges.

2. While the framework is innovative, the paper lacks a detailed analysis of its computational complexity and scalability, which are crucial for practical applications, especially with large datasets. Even more, the training samples are sampled from a look back windows, which means that the effectiveness of the module relies heavily on the training data, potentially affecting the framework's performance in real world data. And how is it able to learn the preicise multiscale statistics predictions without sampling from a multiscale look back window.

3. The paper needs a much more detailed visualization on the learned statistics predictions to prove that the module actually works.

**Questions:**

As is commented in Weakness.

---

> ### Author Response · Authors · 2024-11-21
>
> - Weakness 1:
>
> We conclude potential reasons for method failure as follows: 1) In the online forecasting setting, the multi-scale statistics prediction module is updated only once for each incoming new data, which may challenge the module in capturing some fast-changing statistics and will result in bad performance. Besides, the global dominant periodicity is determined according to the training data, where the relatively small training data split ratio in the online setting may cause an inappropriate periodicity extraction and thus affect the effectiveness of multi-scale slice-based analysis. 2) In the offline forecasting setting, the multi-scale statistics prediction module is only updated on the training data, where the distinct statistics evolving dynamics of training data and testing data may cause a worse performance in the evaluation stage. Despite the potential shortcomings identified above, it is essential to emphasize the overall strengths of the proposed method.
>
> - Weakness 2:
>
> **To analyze the computational complexity**, we have documented the computation time of each part throughout the whole procedure in Table 8 and Table 9. The proposed method will cause extra computational complexity and running time from two perspectives. **First**, the proposed method requires training the multi-scale statistics prediction module, where the complexity is the number of considered periodicities times the complexity of individual prediction modules for a single periodicity. The computational complexity of this part is independent of the complexity of the forecasting backbone model and we have designed this part to be lightweight with a two-layer MLP to avoid a large computation burden. **Second**, the backbone forecasting model will handle the series from multi-scale perspectives, where the complexity is the number of considered periodicities times the complexity of the backbone forecasting model. This part becomes the main computation burden when the backbone model is much more complex than the statistics prediction module. There are **two possible approaches to accelerate the computation process**, one is to let the multi-scale analysis in parallel (our experiment is conducted in series); another approach is to consider an attentive updation strategy in the online learning process, which is to update model only when severe distribution shift is detected instead of updating for every sample. Overally speaking, the computational complexity of our proposed method is acceptable in many real-world applications, especially for those low-frequency forecasting scenarios, such as day-ahead electricity load forecasting.
>
> **To evaluate the scalability of the proposed method**, we have already demonstrated the effectiveness of the proposed method on the largest traffic dataset [1] in Table 6 (where online forecasting in the year 2019 is conducted). We would like to strengthen this statement further by reconducting the experiment on this dataset and **spanning a longer period (5 years, from 2017 to 2021)**, where results again verify the robustness of the proposed method.
>
> |||Online_LSTM||
> |-|-|-|-|
> |||w/o EvoMSN|**+EvoMSN**|
> |Pred_len 48|MAE|35.014|**29.652**|
> ||RMSE|52.289|**43.668**|
> |Pred_len 72|MAE|35.445|**30.104**|
> ||RMSE|53.029|**44.854**|
> |Pred_len 96|MAE|36.868|**29.317**|
> ||RMSE|54.645|**44.053**|
> |Avg|MAE|35.776|**29.691**|
> ||RMSE|53.321|**44.191**|
>
> **For your concern about training data**, the proposed offline-online optimization strategy not only uses samples from the training data but also uses continuous data streams from the testing data to update models, where the effectiveness is validated on datasets that all from the real world. To achieve accurate multiscale statistics predictions, we have sliced the lookback and horizon windows into slices of different lengths and calculated slice statistics, where the slice statistics in the lookback windows serve as inputs to predict slice statistics in the horizon windows.
>
> [1] Largest: A benchmark dataset for large-scale traffic forecasting
>
> - Weakness 3:
>
> Thanks for your constructive suggestions. We have plotted the prediction of statistics from a multi-scale view on the ETTh2 dataset in the uploaded **supplementary material** Fig.1, which shows the prediction module can capture the overall trend of the mean and standard deviation. We find the standard deviation is more volatile and more difficult to predict than the mean, and our future work will investigate more advanced methods to better predict these statistics.

---

> ### Comment · Reviewer_KoUF · 2024-11-26
>
> Thank you for your detailed reply. I would like to maintain my score.

---

### Official Review · Reviewer_SkEV · 2024-11-03

**Soundness:** 3
**Presentation:** 3
**Contribution:** 3
**Rating:** 6
**Confidence:** 3

**Summary:**

This paper introduces a model-agnostic Evolving Multi-Scale Normalization (EvoMSN) framework to tackle the forecasting of
time series with distribution shifts since existing implementations lack the capability of capturing the distribution dynamics
from various scales and modeling the evolving normalized input-output mapping functions caused by gradual distribution
shifts.

The contributions are as follows:

1. EvoMSN is a model-agnostic online normalization framework that enhances any arbitrary backbone forecasting models by
adaptively removing and recovering dynamical distribution information.

2. A multi-scale statistics prediction module is introduced to estimate the statistics of future distributions. An adaptive
ensemble strategy is designed to ensemble the denormalized outputs based on the weights of the local amplitude.

3. An evolving bi-level optimization strategy, including offline two-stage pretraining and online alternating learning, is
proposed to update the statistics prediction module and the backbone forecasting model collaboratively.

4. The effectiveness of the proposed method in boosting forecasting performance under distribution shifts is evaluated on
five large-scale real-world time-series benchmark datasets.

**Strengths:**

The proposed novel framework (EvoMSN) is unique since it combines multi-scale statistics prediction, adaptive ensembling, and
an evolving bi-level optimization strategy to tackle the distribution shift problem in time-series forecasting. It differs from existing
implementations that lack of the capability of capturing the distribution dynamics from various scales and modeling the evolving
normalized input-output mapping functions caused by gradual distribution shifts.

The research is based on sufficient analysis of related existing works. Each component of the proposed framework is well-
introduced. The effectiveness is demonstrated using five large-scale real-world time-series benchmark datasets with appropriate
experiment setup.

The paper is well-organized clear sections for introduction, related works, proposed framework, experiments and conclusion.
The introduction outlines the significance of distribution shift problem in time-series forecasting, the limitations of existing
methods, and the proposed solution.

This paper addresses the distribution shift problem more comprehensively compared to existing implementations. The proposed
framework improves the accuracy of the time-series forecasting. The proposed EvoMSN is model-agnostic that makes it
possible to enhance any arbitrary backbone forecasting models.

**Weaknesses:**

Online forecasting limitations
1. Module update frequency: The parameters of multi-scale statistics prediction module are updated only once for each
incoming new data which might lead to difficulties in extracting some fast-changing statistics, resulting in poor performance.

2. Periodicity extraction: The global dominant periodicity is determined based on the training data. In the online setting, a
relatively small training data split ratio might cause inappropriate periodicity extraction and will affect the effectiveness of
multi-scale slice-based analysis.

Offline forecasting limitations

1. Module update scopes: In the offline pretraining stage, the multi-scale statistics prediction module is only updated on the
training data. The distinct statistics evolving dynamic of training data and testing data may cause worse performance in the
evaluation stage.

Modeling limitations

1. Incomplete distribution characteristics: The multi-scale statistics prediction module only produces the mean and the
deviation of each slice. More comprehensive characteristics of distribution, such as minimum and maximum value, are
significant but neglected in the proposed framework.

**Questions:**

1. The multi-scale statistics prediction module is a two-layer perceptron network. Have you ever attempted to employ any
other different architecture and analyze the sensitivity?

2. Windowing functions are often coupled with FFT. Have you ever attempted to use different windowing functions and
analyze the sensitivity?

3. FFT often assumes the time-series data is stationary. When this assumption is violated, the resulting frequency-amplitude
information might be not meaningful. Have you consider using any other approaches, and if so, how does the performance
vary?

---

> ### Author Response · Authors · 2024-11-21
>
> Thank you for your acknowledgment and valuable feedback on our work, we would like to address your concerns as follows.
>
> - Weakness:
>
> Thanks for re-stating the limitation of the proposed method that we have analyzed in Appendix B. We will further work on these perspectives to improve the technique in our future work.
>
> - Question 1:
>
> Thank you for your insightful comments and for drawing attention to the architecture of our multi-scale statistics prediction module. In response to your question, we have indeed considered various architectures for our prediction module. Initially, we experimented with different neural network configurations, including convolutional neural networks (CNNs) and recurrent neural networks (RNNs), to assess their effectiveness in capturing time-series dependencies and patterns. However, after a series of experiments and cross-validations, we found that the two-layer perceptron network (MLP) provided a good balance between model complexity and predictive accuracy for our specific dataset and task. The MLP's simplicity allowed for faster training times and easier interpretation of the model's weights, which were crucial factors for our research. We provide visualization of prediction of statistics from a multi-scale view on the ETTh2 dataset in the uploaded **supplementary material** Fig.1, which shows the proposed prediction module can capture the overall trend of the mean and standard deviation. We find the standard deviation is more volatile and more difficult to predict than the mean, and our future work will investigate more advanced methods to better predict these statistics.
>
> - Questions 2 & 3:
>
> We appreciate your concern regarding the potential limitations of FFT for time series analysis. The main reason for us to apply FFT is its efficiency in extracting multi-periodicity information from time series, which is also adopted by various mainstream studies [1,2]. We mainly utilize FFT from a global perspective to determine the dominant periodicity to guide the window slicing. Our design considers ensembling multiple outputs according to their local periodicity amplitude, which can mitigate the adverse effects of inappropriate periodicity extraction and thus make our forecasting more robust. We will investigate more adaptive window slicing (such as methods based on changepoint detection) and multi-scale modeling strategies in our future work.
>
> [1] Wu H, Hu T, Liu Y, et al. Timesnet: Temporal 2d-variation modeling for general time series analysis[J]. arXiv preprint arXiv:2210.02186, 2022.
>
> [2] Zhou T, Ma Z, Wen Q, et al. Fedformer: Frequency enhanced decomposed transformer for long-term series forecasting[C], International conference on machine learning. PMLR, 2022: 27268-27286.

---

> > ### Comment · Reviewer_SkEV · 2024-11-22
> > **Response to Rebuttal**
> >
> > I have reviewed the authors' rebuttal, and while they have addressed most of my concerns, I will maintain my current score due to the paper's quality, novelty, and presentation.

---

### Official Review · Reviewer_7Pz9 · 2024-11-06

**Soundness:** 2
**Presentation:** 3
**Contribution:** 3
**Rating:** 5
**Confidence:** 2

**Summary:**

This paper introduces the Evolving Multi-Scale Normalization framework, a model-agnostic approach designed to address complex distribution shifts in time series forecasting. EvoMSN combines multi-scale statistics prediction, adaptive ensembling, and an evolving bi-level optimization strategy to improve the accuracy of long-term forecasts. The framework is evaluated on several real-world datasets and shown to significantly enhance the performance of five mainstream forecasting methods.

**Strengths:**

1. The EvoMSN framework introduces the concept of multi-scale statistics prediction, extending existing normalization methods. By combining this with adaptive ensembling and online learning strategies, the framework better captures and responds to data distribution changes. This integration of multiple techniques provides a new approach to addressing distribution shifts in time series forecasting.
2. The authors present robust experimental results that demonstrate the effectiveness and superiority of EvoMSN. These experiments span a variety of real-world datasets and utilize multiple advanced forecasting methods as backbones, ensuring the reliability and generalizability of the findings.
3. The paper is well-structured and logically coherent.

**Weaknesses:**

1. The paper aims to tackle the complex distribution shifts problem, but the description of what constitutes these distribution shifts is somewhat vague. It would be beneficial to provide a more specific description of how the distributions change. For instance, among the distributions \(P(X)\), \(P(Y)\), \(P(X|Y)\), and \(P(Y|X)\), which ones are changing and which ones remain constant? A clearer delineation of these changes would enhance the understanding of the problem being addressed.

2. The paper lacks a detailed explanation of why normalization and denormalization can effectively address distribution shifts in time series forecasting. Providing a theoretical or empirical justification for this approach would strengthen the paper's arguments and help readers understand the underlying mechanisms better.

3. The paper states that the EvoMSN framework can be combined with existing time series forecasting methods to improve their performance. However, it does not quantify the increase in computational complexity during training and testing when incorporating the EvoMSN framework. Understanding the trade-off between performance improvement and computational cost is crucial for practical applications.

4. Section 3.3 introduces an offline two-stage pretraining and online alternate updating method to solve the bi-level optimization problem. The paper does not adequately explain why this particular method was chosen and whether there are theoretical guarantees for the optimization results obtained using this approach. A more detailed discussion of the advantages and potential limitations of this method would be beneficial.

5. The experimental results show that the EvoMSN framework does not consistently improve the performance of the models. The paper does not provide a detailed analysis of why this inconsistency occurs. A thorough investigation of the conditions under which the framework performs well or poorly would help in understanding the robustness of the proposed method and guide future improvements.

**Questions:**

Refer to the weakness.

---

> ### Author Response · Authors · 2024-11-21
>
> We greatly appreciate your acknowledgment of our proposal. We would like to address your concerns as follows.
>
> - Weakness 1:
>
> Denote the distribution of a lookback window and a horizon window at timestep $t$ as $P_t(X)$ and $P_t(Y)$, respectively. The conditional distribution that reflects the input-output mapping relationship is denoted as $P_t(Y|X)$. First, the shifts in marginal distribution happen between the lookback and horizon window $P_t(X) \neq P_t(Y)$ and also between the window of a different timestep $P_t(X) \neq P_{t_n}(X), P_t(Y) \neq P_{t_n}(Y)$. Second, the conditional distribution shifts all the time $P_t(Y|X) \neq P_{t_n}(Y|X)$, especially between the training and testing data $P_{train}(Y|X) \neq P_{test}(Y|X)$. These distribution shifts are coupled and happen simultaneously.
>
> - Weakness 2:
>
> The normalization approach will remove the dynamic distribution information to let the backbone forecasting model better learn from the data with a stable distribution, while the denormalization process restores the distribution information to the outputs. The key to addressing distribution shifts is to accurately model how the distribution is shifting in order to facilitate time series forecasting. The empirical justifications are shown in Table 1, Fig.6, and Table 13, which show that models' performance will be degraded by distribution shifts without the proposed adaptive normalization approach.
>
> - Weakness 3:
>
> We have documented the computation time of each part throughout the whole procedure in Table 8 and Table 9. The proposed method will cause extra computational complexity and running time from two perspectives. **First**, the proposed method requires training the multi-scale statistics prediction module, where the complexity is the number of considered periodicities times the complexity of individual prediction modules for a single periodicity. The computational complexity of this part is independent of the complexity of the forecasting backbone model and we have designed this part to be lightweight with a two-layer MLP to avoid a large computation burden. **Second**, the backbone forecasting model will handle the series from multi-scale perspectives, where the complexity is the number of considered periodicities times the complexity of the backbone forecasting model. This part becomes the main computation burden when the backbone model is much more complex than the statistics prediction module. There are **two possible approaches to accelerate the computation process**, one is to let the multi-scale analysis in parallel (our experiment is conducted in series); another approach is to consider an attentive updation strategy in the online learning process, which is to update model only when severe distribution shift is detected instead of updating for every sample. Overally speaking, the computational complexity of our proposed method is acceptable in many real-world applications, especially for those low-frequency forecasting scenarios, such as day-ahead electricity load forecasting.
>
> - Weakness 4:
>
> We design this specific offline-online optimization strategy for the following motivations: The offline two-stage training process facilitates the statistics prediction model to focus on capturing distribution information and the backbone model to focus on learning from stationarized data. The offline stage enables models to learn from the past and provides a generalizable starting point for subsequent online learning. The online stage helps models track evolving distributions by learning from the continuous data stream. Specifically, the alternative updation strategy aims to optimize the lower and upper objectives sequentially and avoids over-fitting problems caused by blending the training of all components. The potential limitation of the strategy is that it may encounter the plastic-elastic dilemma and forgetting problem during the online learning stage, which will be further investigated in our future work.
>
> - Weakness 5:
>
> We conclude potential reasons for method failure as follows: 1) In the online forecasting setting, the multi-scale statistics prediction module is updated only once for each incoming new data, which may challenge the module in capturing some fast-changing statistics and will result in bad performance. Besides, the global dominant periodicity is determined according to the training data, where the relatively small training data split ratio in the online setting may cause an inappropriate periodicity extraction and thus affect the effectiveness of multi-scale slice-based analysis. 2) In the offline forecasting setting, the multi-scale statistics prediction module is only updated on the training data, where the distinct statistics evolving dynamics of training data and testing data may cause a worse performance in the evaluation stage. Despite the potential shortcomings identified above, it is essential to emphasize the overall strengths of the proposed method.

---

### Note · Authors · 2024-12-28

I have read and agree with the venue's withdrawal policy on behalf of myself and my co-authors.